# Entity Linking for real-time geolocation of natural disasters from social network posts

**Gaëtan Caillaut** [1,4]☯*, **Samuel Auclair** [1]☯, **Cécile Gracianne**[1], **Nathalie Abadie**[2], **Guillaume Touya**[3]

**1** BRGM, Department of Risks and Prevention, BRGM, Orléans, France, **2** LASTIG, Univ Gustave Eiffel, IGN-ENSG, Saint-Mandé, France, **3** LASTIG, Univ Gustave Eiffel, IGN-ENSG, Champs-sur-Marne, France, **4** Lingua Custodia, Paris, France

☯ These authors contributed equally to this work.
* gaetan.caillaut@linguacustodia.com

**Data Availability Statement:** ***PA AT ACCEPT: Please follow up with authors to upload data*** The data underlying the results presented in the study are the property of Twitter. While we have done our best to release the data used, we cannot

## Abstract

When a fast kinetic natural disaster occurs, it is crucial that crisis managers quickly understand the extent of the situation, especially through the development of "big picture" maps. For many years, great efforts have been made to use social networks to help build this situational awareness. While there are many models for automatically extracting information from posts, the difficulty remains in detecting and geolocating this information on the fly so that it can be placed on maps. Whilst most of the work carried out to date on this subject has been based on data in English, we tackle the problem of detecting and geolocating natural disasters from French messages posted on the Twitter platform (now renamed "X"). To this end, we first build an appropriate dataset comprised of documents from the French Wikipedia corpus, the dataset from the CAp 2017 challenge, and a homemade annotated Twitter dataset extracted during French natural disasters. We then developed an Entity-Linking pipeline in adequacy with our end-application use case: real-time prediction and peak resiliency. We show that despite these two additional constraints, our system's performances are on par with state-of-the-art systems. Moreover, the entities geolocated by our model show a strong coherence with the spatiotemporal signature of the natural disasters considered, which suggests that it could usefully contribute to automatic social network analysis for crisis managers.

## 1 Introduction

Crisis situations are characterised by a collapse of meaning and by the disorientation of actors. Crisis management, therefore, consists, first and foremost, in re-establishing this meaning by understanding the situation and building "situational awareness". As this "sensemaking" process is a necessary prerequisite for the construction of an appropriate response, it takes place under a strong time constraint and therefore requires the activation of all information channels that can be used to gather reliable elements that describe the situation as quickly as possible [1]. In addition to traditional actionable channels, which often take time to gather consolidated information from the field, it has become common practice over the past 10

guarantee its integrity nor its general availability due to the Twitter terms of services. For instance, some tweets may be deleted or the Twitter API may change. The training dataset used to train the models is publicly available on Zenodo at https://doi.org/10.5281/zenodo.7767294 and https://doi.org/10.5281/zenodo.7767294. The model used in this work will be released after publication of this work. For now, reviewers can access the code and the weights of the model on https://figshare.com/s/3d5faa5258d1346dbe01.

**Funding:** The work presented in this article was carried out with funding from the French Research Agency (ANR - https://anr.fr/en/) within the framework of the CARNOT institutes (Gaëtan Caillault), as well as within the the RéSoCIO project co-funded by ANR under the grant ANR-20-CE39-001 (Samuel Auclair & Cécile Gracianne). Opinions expressed in this paper solely reflect the authors' view; the ANR is not responsible for any use that may be made of information it contains. The funders had no role in study design, data collection and analysis, decision to publish, or preparation of the manuscript.

**Competing interests:** The authors have declared that no competing interests exist.

years for crisis practitioners to try to capture the information circulating on social media. Among all social media platforms, Twitter (now renamed "X") proposes particularly useful features in a monitoring crisis situation: publication in real time of short messages, streaming API that makes it possible to automate monitoring tasks (free until the beginning of 2023, the Twitter / X streaming API is now fee-based, with exemptions granted to certain applications broadcasting emergency notifications), ability to join images, etc. Although Twitter is not the most widely used social media, its user community remains significant: in the case of France, which is considered in this study, the number of active users is estimated at around 10 million i.e., almost 15% of the total population. Therefore, the occurrence of natural disasters often results in massive and immediate spread of tweets [2] that leads to the consideration of the Twitter platform as valuable "Distributed Sensor System" [3]. In practice, the richest information generally comes from the citizens closest to the disaster area. Because natural disasters affect their immediate environment, these *local citizens* [4] are indeed more inclined, both in the physical and digital sphere, to help or exchange objective information about the situation on the ground [5]. Thus, after the occurrence of natural disasters, many users of social networks concerned by the situation massively exchange information concerning the intensity of the events as they perceive them: either via a description of the phenomena themselves (severity of earthquakes, the extension of flooded areas, wind strength, etc.) or via a description of their effects (damage to buildings and infrastructure, shutdown in transport, energy or telecommunications networks, victims, etc.).

In practice, social network monitoring in support of crisis management is often carried out by dedicated Virtual Operations Support Teams (known as "VOST") [6]. In the absence of dedicated tools, VOST members usually carry out this monitoring work "manually" via the public search interfaces provided by platforms. From this perspective, it would be particularly useful to develop tools able to automatically extract actionable information from large quantities of messages: VOST members could then focus on validating and analyzing the automatically extracted information, rather than wasting time extracting this information on their own message by message. One of the main issues of these automatic analyses is the ability to correctly place the information extracted on a map to make it actionable [7] and thus reduce information overload. Because a very low proportion of tweets have intrinsic geolocation (less than 1%), such as GPS coordinates [8, 9] or geotags [10], geolocation of the information should be inferred from the information present in the text and/or the metadata of the tweet itself.

In view of supporting crisis management practitioners in France and in French-speaking countries, the *SURICATE-Nat* platform (https://www.suricatenat.fr/) allows for the continuous monitoring and analysis of original tweets (i.e. excluding retweets) written in French after the occurrence of natural disasters (earthquakes and floods) [11]. Each tweet captured thanks to the Twitter streaming API is processed to extract thematic information, as well as a preferred geolocation. The geographic inference module implemented on the *SURICATE-Nat* platform is based on a *Named Entity Recognition* (NER) tool developed natively to handle the French language [12], supplemented by the use of OpenStreetMap Web services to obtain the coordinates of recognised geographic entities. Although this approach has shown its usefulness by detecting relatively well the mentions of French municipalities in the event of a natural disaster [13], it often fails and does not allow one to recognise points of interest [11].

Meanwhile, recent advances in natural language processing (NLP) and deep learning allowed significant improvements in the recognition and disambiguation of spatial named entities using geographic coordinates prediction, geographic region classification, gazetteer-based approaches, or, as explored in this work, entity-linking (see [14] for a review). Entity linking gave very promising results in the prediction of entities from knowledge bases such as Wikipedia/Wikidata [15]. In particular, this approach comes out to be effective in zero-shot

classification, hence allowing the prediction of accurate locations of places that did not yet exist at training time. Furthermore, precise geographic coordinates of spatial entities can be extracted from Wikidata, as well as many other features, such as OpenStreetMap identifiers. However, using this strategy requires appropriate corpora to train models, which can be challenging for applications based on non-English-speaking social media data.

In this context, our key contributions are threefold: we first extend the work done by [14] by producing a French Entity-Linking dataset covering (most of) the French Wikipedia corpus as well as annotated tweets collected during major crises in France. Then, our main contribution is an innovative pipeline to geolocate places impacted by natural disasters from tweets automatically. This pipeline implements the entity-linking framework, while unusual, it fixes lots of precision issues traditional geolocation systems. It relies on a custom bi-encoder that simultaneously recognises **all** spatial named entity mentions in a text and generates entity candidates from Wikidata at **the same time**, while existing EL pipeline usually rely on two different models processing one entity at once. We also rely on an optimized cross-encoder to rerank the entity candidates. Geolocation is then done by querying spatial information, such as GPS coordinates, from the predicted Wikidata entities. This model avoids the shortcomings of the methods from the literature: spatial (im)precision, the ambiguity between spatial entity names, or the need for a specific gazetteer, and the scores obtained on the Entity Linking task are better than the literature. Codes and models are available online (https://github.com/gcaillaut/geoloc-entity-linking). The last contribution is a geography-oriented qualitative evaluation method for the problem of geo-locating tweets about natural disasters. We applied this method on two datasets of tweets emitted during two natural disasters and found out that the model predictions can be used to have a clear spatiotemporal description of each event.

## 2 Task description and related works

The geolocation problem that we are considering consists in assigning coordinates or a geometry (point, polyline, or polygon) described as a set of coordinates expressed in a well-known coordinate reference system to a civic location (e.g. a place name or a geographic identifier such as a post-code) captured via social media and, more particularly, via the Twitter platform. Consequently, it is necessary first to define precisely which place we are trying to locate. In fact, there are methods to locate the place where tweets are sent, the place of residence of their sender, or the place(s) mentioned in the tweets [16]. As our goal is to capture the maximum amount of information describing the situation related to the ongoing natural disaster, it is clear that the important thing is to localise the information contained in the messages (i.e., potentially several civic locations per tweet), which corresponds to a "mentioned location prediction", or geoparsing, problem [16]. Although the locations mentioned in a tweet regularly differ from the location of the user [17], we are interested in tweets from both direct and indirect witnesses of the disaster. In addition, [18] showed that there is a strong correlation between the locations mentioned in tweets and the location of their author in natural disaster situations. Several different methods have been proposed in the past ten years to solve this location prediction task on Twitter [16, 19], and this section does not have the ambition to discuss them all, but we will focus more on methods addressing a similar crisis management context. Then, we present how entity linking can be used to predict locations and lastly, we review the available training datasets.

### 2.1 Location prediction with tweets in crisis management context

Many of the first attempts modelled this task as a grid-based geolocation problem, where a grid is superimposed on the geographic space and the geolocation consists in inferring which

cell or portion of space is referred to in the tweet (e.g. [20]). We voluntarily leave these approaches aside because they predict locations whose planimetric accuracy is not sufficient for our use case; i.e. the cells are too large, or it is too complex when cells are small. We are interested in methods that are able to infer more precise geolocations from the tweets, to really help the crisis management. Hybrid approaches have been developed to overcome this limitation, but even the best ones cannot reach the level of precision expected, and required, in our context. Indeed, places are often geolocalised within a margin of multiple hundred kilometres. Recent approaches have achieved better mean and median distance scores by feeding deep learning models with more spatial knowledge like geophysical information about cells. Their results still fall short of those obtained by approaches based on gazetteers [21]. Furthermore, these approaches tend to output raw coordinates, which are usually not meaningful to people. For instance, the coordinates (48.86382, 2.30202) are probably not meaningful to most people, while referring to "the Zouave" (https://en.wikipedia.org/wiki/Zouave_(Pont_de_l'Alma)), the famous statue located on the Alma Bridge in Paris is. We hence need a method that can both predict precise coordinates and associate them with familiar place names. An effective way to achieve that is to geolocate geographic mentions in texts through Entity Linking. This approach has been adopted in many tweets geolocalization projects, whether for messages dealing with natural disasters or not, for example [22]. These works differ from one another in the approaches implemented to solve the tasks of recognizing spatial named entities and linking recognized mentions.

Usually, the first task is to recognise the spatial named entities. In the literature on practical applications of tweet content geolocation for crisis or emergency management, this recognition is a matching task between elements in a spatial, sometimes hierarchical, gazetteer and the words in the text [23–28]. There are two main problems with this frequently used matching approach: (1) some names are ambiguous because used to describe several geographic features, and the ambiguity is even higher with the language used in tweets; (2) the gazetteer might not contain all the place names required to geolocate the tweets. For instance, some methods require prior knowledge of the region of the event to collect the appropriate gazetteer [26], which is not possible for us. Based on the observation that there is little training data available for recognising and disambiguating spatial named entities in tweets, especially in a multilingual context, [29] investigates several training strategies on a BERT-based language model to improve spatial named entities recognition in tweets dealing with natural disasters. [30, 31] match place names extracted from tweets by BERT-based pre-trained models, with a gazetteer (GeoNames and OpenStreetMap), which does not require any manual annotation step.

Although the named entity recognition task can be complex with the size and language used in tweets, location inference disambiguation is considered the most difficult task [24, 32]. [33] uses four existing semantic annotation tools, namely DBPedia spotlight, TagMe, Dexter 2.0 and Dandelion, to recognize place name mentions in tweets dealing with natural disaster events. Coordinates are then retrieved by the system using spatial properties associated with the gazetteer entries provided by the semantic annotators, and their equivalent resources. A SVM classifier is finally trained to decide which coordinates are the best for each mention, based on criteria such as textual features (POS tags or case), string similarity between mentions and gazetteer labels, etc. This approach combines the results of several semantic annotation tools to select only the best. As such, it highly depends on the performance of the chosen semantic annotation tools, and especially on their ability to process a given language. Rather than just using the text content of the tweet, the strategy proposed by [34] compares the content of the tweet to the others and uses a majority vote based on the most similar tweets to disambiguate the mentioned locations, and that described in [35] proposes to use the Twitter user social network. Others use the general context of the tweet, such as the history of the

user's tweets [27, 36]. Though these approaches are theoretically interesting, they are not applicable to us because we target a real-time geolocation inference while parsing the history of the social network is computationally intensive. Moreover, the two latter strategies tend to locate the place where Twitter users live, not the tweet content itself. The strategy proposed by [37] uses both tweet's textual and image content. Place names are recognized in tweets textual content using NeuroNER, a LSTM model for natural language processing. Their geographic coordinates are then retrieved by matching place name mentions with toponyms from a gazetteer built from GeoNames and US Census data TIGER (road network). Tweets are then filtered by keeping only those containing images classified as showing "Impact" flooding phase events and referring to road-level locations as they are supposed to be the most useful for emergency services. This approach seems relevant to filter tweets that are very relevant. Still, the named entity mentions linking step with the gazetteer is not described in detail and may be based on a simple comparison of character strings. In the case of homonyms, only the presence of a photograph classified as representing a natural disaster can distinguish between two candidate locations. Another strategy for this disambiguation problem is to use knowledge graphs that describe the semantic proximity of the named entities [25]. In this last paper, the knowledge graph is very detailed, but created in a small region, which is not possible for us as we target geolocation in the whole French territory. Building such a detailed resource on the whole French territory would be very costly. But providing the model with knowledge about the type of named spatial entities to be located, their neighbourhood (nearby points of interest likely to be mentioned) and their type of environment (urban, rural, etc.) is a promising avenue. Lastly, [38] analyses tweets sent by people calling for help during Harvey hurricane.As people were trying to reach the emergency services for help, they wrote down their addresses. This study therefore aims to extract a very specific category of named spatial entities in tweets: addresses. The approach adopted is based on rules, because addresses are highly structured named entities. They are then located using the Google Geocoding API. This is a very specific use case which does not reflect the diversity of geographic entity types mentioned in tweets about natural disasters.

One final issue is the visualisation of the locations mentioned in the tweets in order to help experts browse and analyse the information from the tweets [39]. This issue is not addressed in this article.

## 2.2 Location prediction as an Entity Linking task

The Entity Linking (EL) task consists in linking an *entity mention* found in a document to its corresponding *entity* in a Knowledge Base (KB). The notion of entity mention has to be distinguished from the notion of entity: an entity is a unique and normalized instance corresponding to an entry in the targeted KB, while an entity mention is a small chunk of text in a document referring to an entity. For instance, `Paris` is the entity corresponding to the capital of France and "Paris", "City of light" or "capital city of France" are mentions referring to it.

Entity Linking often relies on three sub-tasks: mentions detection, candidate generation and candidate filtering (or ranking).

The *mentions detection* (MD) task consists in detecting the entity mentions, that is to say, extracting spans of text referencing entities. For instance, given the sentence "Paris is the capital of France", a mention detector could extract three mentions: "Paris", "capital of France" and "France". This task is very similar to the NER task, where mentions are detected and classified. Typically, a NER classifier would classify the three previous entity mentions as `GEOLOC` entities.

The MD task is critical in the EL pipeline since mentions must obviously have been detected prior to the linking step. Fortunately, approaches tend to solve the NER and MD tasks extremely efficiently by fine-tuning large pre-trained language models on a token classification task, where token labels are usually following the `BIO` scheme, where starts of mentions are labelled with `B-X` labels (X being the type of the entity), in-mention tokens with `I-X` labels and other tokens with `O`. However, neither nested (as in the previous example) nor discontinuous entity mentions can be detected in this way due to the limitations of the `BIO` format. Naturally, this theoretical limitation has been tackled by many researchers [40, 41] while, in practice, this limitation does not really impact EL models, since the main dataset for training EL systems, Wikipedia, does not contain these types of annotations.

*Candidates generation* relies on the previous step to generate a set of potential entity candidates for each detected entity mention. For example, the mention "France" can refer to `France`, the country, or `France Gall`, the famous French singer. Most recent approaches to candidates generation tend to expect a sentence with a single annotated mention as input [15, 42, 43], which means that sentences containing multiple entity mentions must be fed multiple times in the system, thus decreasing the throughput of the overall EL pipeline. An exception is the GENRE model [44] which relies on a sequence-to-sequence architecture to enrich the input sentence with annotations about the mention and the entity.

Finally, the *candidates filtering* step aims to accurately filter, or rank, the set of candidates previously generated. This step generally requires much more computational power than the *candidates generation* step and hence cannot be used to rank the whole set of entities in the considered KB, which is usually very large.

## 2.3 Training corpora for spatial named entity linking

Training an EL system requires a corpus annotated with both entity mention spans and target entities. Few gold standard datasets are available, and the ones that are released are outdated, hence missing many recent entities. For example, both the AIDA CoNLL-YAGO dataset [45] and the TAC 2010 datasets are more than 10 years old as of writing this paper. Thus, a model trained on these datasets probably cannot capture the current state of the world, or only partially. For example, French's administrative regions have been reformed in 2016. Since those changes are posterior to the training dataset creation, a model trained on it would probably not be able to link a mention of a new region to the right entity. The same reasoning applies to new facilities that can be used to anchor an event to a location. Although this may not be dramatic in most applications, incorrectly or not predicting a location could have many severe repercussions in the application we are targeting in this work.

As a consequence, most works rely on crowd-sourced datasets that are continuously updated and maintained by people all over the world. Wiki-like corpora have gained a lot of interest in the EL community since users are spontaneously and massively writing and annotating wiki pages, where mentions are identified by links to other wiki pages and target entities correspond to the target pages. For instance, [15] are relying on WikiNews (https://www.wikinews.org/) and [42] on Fandom (https://www.fandom.com/) (formerly called Wikia).

However, such crowd-sourced datasets are often written in a formal language level differing from the level of language used on social networks, making it difficult to train a model that fits the posts on social networks. Furthermore, such documents lack crisis-related annotations, such as assessments of damages, which are extremely valuable in the event of a natural disaster. Some crisis-related corpora have been extracted from Twitter [46, 47], unfortunately, they generally target exclusively English tweets, making them unusable to deal with a French crisis.

## 3 The Entity-Linking framework

Detecting and geolocating the effects of natural disasters must be done in real time to provide useful and up-to-date insights to crisis managers and emergency teams. As such, our work has been guided by the multiple requirements of our application context: low latency, since the situational knowledge must not be outdated, and resiliency to peaks of data reception, since the starts of events are often correlated with a surge of tweets emission.

While entity-linking has already been studied for a long time [48–52], it is quite uncommon to see this approach applied on the automatic geolocation task. Yet, most geolocation methods are not precise enough to be used during emergencies, while entity-linking allows the mapping of a given text mention to a unique entity, from which on can extract precise coordinates or geographic region. In this section, we will describe our own original approach addressing the aforementioned requirements using an original and efficient entity-linking pipeline composed of a two-in-one bi-encoder detecting and classifying mentions, and an optimized cross-encoder trained on realistic data.

Modern approaches to solving the EL task tend to expect a document where a single mention to link to the target knowledge base has already been annotated, supposedly by a NER system. This requires first processing the input document to detect entity mentions and then feeding these annotated mentions to the EL system. A significant drawback of this approach is that detecting mentions beforehand adds a serious overhead to the whole EL process, especially considering that both the MD and EL models tend to rely on similar large pre-trained language models.

Furthermore, few works propose a solution to handle multiple mentions in the same document. As a consequence, documents containing several mentions need to be duplicated and each mention must be processed separately, which necessarily has a negative impact on the whole EL process throughput.

In this work, we propose a novel approach mitigating the overhead of modern EL systems by **not relying on a prior mentions detection step** and by **enabling linking of multiple mentions simultaneously** at the same time. Our framework is composed of two main components: (1) a bi-encoder that encodes in parallel the geo-location mentions, and the entity definition from Wikipedia (see Section 3.1), and (2) a cross-encoder that links the detected mentions with the good entity definition from Wikipedia (see Section 3.2).

### 3.1 Multi-tasks bi-encoder

The bi-encoder architecture [15, 53] has been designed to map inputs of different nature to the same latent space. It relies on two different encoders (which may share their parameters) trained to produce similar embeddings given different *flavors* of the same object. In this work, we trained a bi-encoder to build similar embeddings for an entity mention and its corresponding Wikipedia definition.

We propose an evolution of the bi-encoder model allowing to, simultaneously, detect entity mentions in an input document **and** producing embeddings for **all of them** in a single forward pass. The overall architecture is illustrated in Fig 1.

The bi-encoder architecture has been designed to map entity mentions and entities to the same latent space. It is compounded of two encoders producing embeddings in parallel. The first encoder is called the *Mention Encoder* and produces mention embeddings given an input sentence. The sentence is processed by a language model, and the resulting word embeddings are forwarded to two small feed-forward neural networks to detect entity mentions (the NER classifier) and to produce their embeddings.

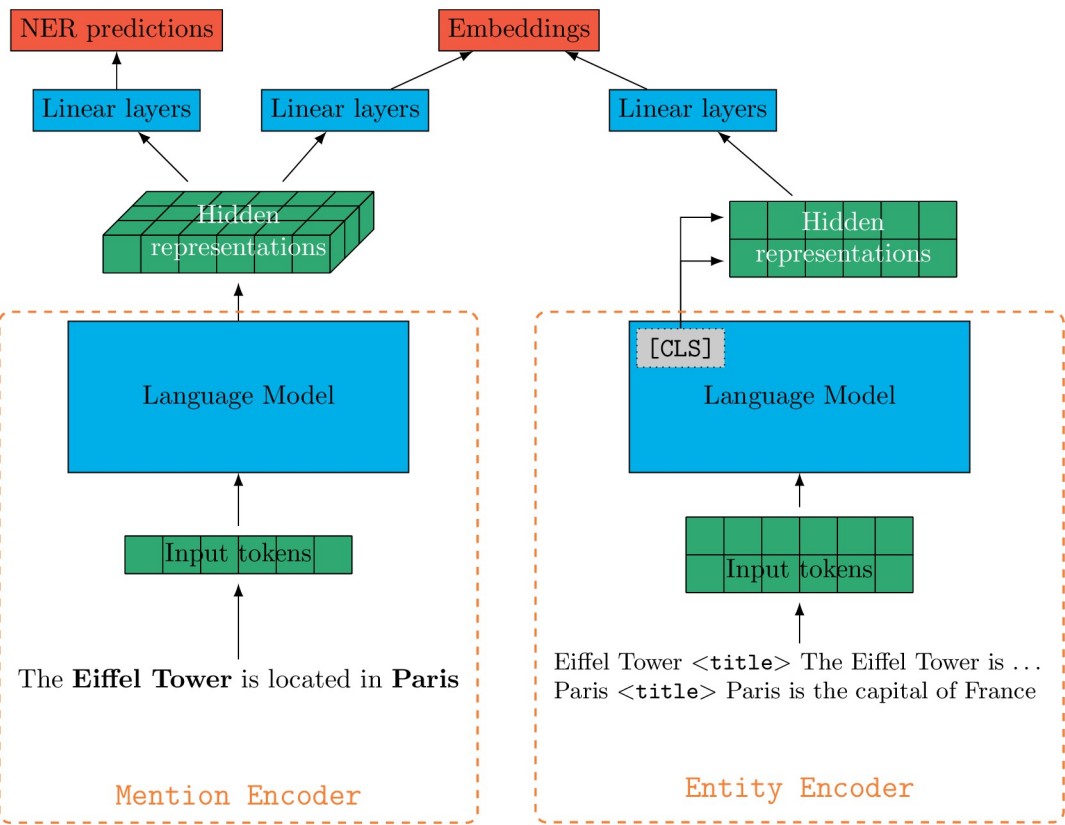

**Fig 1. Architecture of the bi-encoder model.** The *Mention Encoder* processes the input sentence to detect entity mentions (here "Eiffel Tower" and "Paris") and uses the first tokens of the detected mentions to produce embeddings. The *Entity Encoder* is trained to produce embeddings from the descriptions of the entities referenced by the mentions in the input sentence. Both encoders are trained to output similar embeddings.

The second encoder is the *Entity Encoder* which produces entity embeddings given a short description of the entity. We rely on the `[CLS]` token to build a representation of the input entity and then we mapped it to the same latent space as the Mention Encoder thanks to a small feed-forward neural network.

These encoders are trained to output similar representations for the given in-context mentions and their target entity descriptions. As described in [54], Both the Mention Encoder and the Entity Encoder correspond to the first four layers of a CamemBERT model, initialised following the recommendations of [15].

Entity embeddings can be, and should be, pre-computed before inference, by feeding all known entities' descriptions to the Entity Encoder. As such, only mention embeddings need to be computed, by the Entity Encode, during inference. This enable fast and efficient retrieval of candidates by comparing mention embeddings to pre-computed entity embeddings. Similarity between embeddings is often computed with dot product or cosine similarity. In this work, we rely on dot product.

Some works [15] rely on the special `CLS` token to produce an embedding from a single annotated entity mention and its context. We first propose to remove the need to annotate mentions beforehand by enabling the Mention-Encoder to detect mention spans. This is simply done by appending a token classifier head on top of the Mention-Encoder. Then, we propose to rely on the first token, instead of the `CLS` token, of an entity mention to produce

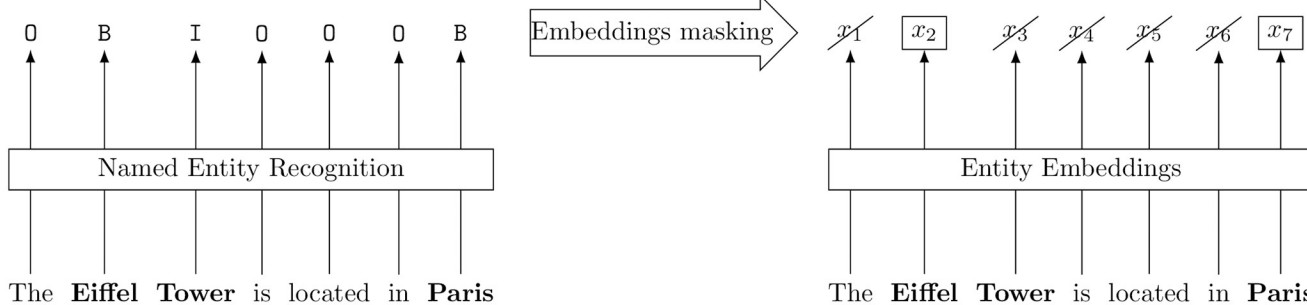

**Fig 2. At inference time, the NER outputs are used to mask the non-entity embeddings produced by the mention-encoder.** Only the first token (B tokens) of each mention is kept and compared to the embeddings pre-computed by the Entity Encoder. For simplicity's sake, we suppose that each word engenders exactly one token.

mention embeddings, thus allowing to embed all the entity mentions of a document at the same time. At training time, since the start position of each mention is known, the output of the NER classifier is not used to select the entity embeddings. However, the output of the NER classifier is crucial at inference time since it is the only way to know the positions of entity embeddings. So, each token embedding classified as B-X (where X is one of the considered entity types) gives rise to an output mention embeddings. Each token embedding classified as B-X is given to the last feed-forward neural network to produce the final mention embedding. This embedding selection process is illustrated in Fig 2. It may be possible to improve our architecture by introducing an additional cross-attention layer merging all the tokens of a given mention into one single mention embedding, but we argue that this cross-attention can already be performed in the previous attention layers of the encoder, without requiring to complexify further the current architecture.

The model is trained to minimize conjointly the loss $\mathcal{L}_{NER}$ associated to the NER task and the loss $\mathcal{L}_{EL}$ associated to the EL task:

$$\mathcal{L}_{NER}(o_{ner}, y_{ner}) + \mathcal{L}_{EL}(m_{el}, e_{el}) \tag{1}$$

Where $o_{ner}$ are the predicted NER labels, $y_{ner}$ are the expected NER labels, $m_{el}$ are the output entity representations from the mention encoder and $e_{el}$ are the output entity representation from the entity encoder. The $\mathcal{L}_{NER}$ function is the classical cross-entropy loss and $\mathcal{L}_{EL}$ is the in-batch cross-entropy loss [55], which is computed on the dot-product between the mention embeddings and the entity embeddings.

After training, the encoders can be separated and the Entity-Encoder should be used to pre-compute embeddings from the descriptions of the set of all known entities. Since the Entity-Encoder has been trained from textual descriptions, it can also produce embeddings of entities not seen during training time in a Zero-Shot learning fashion. Entity embeddings will be compared to outputs produced by the Mention-Encoder to generate a set of candidate entities for each mention detected in the input sentence. However, the true target entity is not necessarily included in the set of candidates, either because the model is wrong or because the target entity does not exist in the target knowledge base. In the first case, the error can be coming from an erroneous NER prediction or from an inadequate mention or entity embedding. The second case could occur when trying to map a mention to an entity that was not in the knowledge base at the time when entity embeddings were pre-computed. To prevent these kinds of errors, we rely on a cross-encoder to compute matching scores between an in-context mention and the selected candidates.

## 3.2 The cross-encoder

A cross-encoder is a language model trained on sentence pairs, allowing to compute cross-attention between tokens of the two input sentences. It is particularly useful to predict a matching score between sentences, as required, for instance, by the Next Sentence Prediction task used to train the BERT model [56]. [42] propose to rely on this architecture to compute similarities between an in-context mention and candidate descriptions.

Our model has been trained on sentence pairs where the first one contains an annotated mention and the second one is the description of a candidate entity. The mention in the first sentence is annotated using special `<start>` and `<end>` tokens to delimit the mention tokens. The second sentence is the concatenation of the target entity name and a textual description, separated by the special token `<title>`. In our experiments we used the Wikipedia titles as entity names and the descriptions are the first sentence of each Wikipedia page, since Wikipedia's guidelines instruct authors to write a short and descriptive first sentence to introduce the page's subject. An input of the Cross-Encoder could hence be:

- in-context mention: "The `<start>`" Eiffel Tower `<end>` is located in Paris.

- candidate description: "Eiffel Tower `<title>`" The Eiffel Tower is a wrought iron lattice tower on the Champ de Mars in Paris, France.

## 4 Data and training materials

Most of the previous works rely on Wikipedia to train Entity-Linking systems. Indeed, Wikipedia offers various benefits: being a large multilingual encyclopedia, Wikipedia is a key resource for many NLP tasks since it is a massive corpus dealing with an extensive set of topics. It is particularly precious for the Entity-Linking task because it is one of the few corpora to have mention annotations and links to their corresponding entities. Mention annotations are materialized by internal links between Wikipedia pages (the `<a>` HTML tag), and their associated entity is the link's target page (the `href` attribute). Moreover, Wikipedia pages are associated with a node in the Wikidata ontology, simplifying the collection of information about a particular entity. For instance, GPS coordinates are available for most of the geographic entities as well as links to gazetteers (such as OpenStreetMap), which is particularly helpful for our target application.

However, models trained on Wikipedia may not be well-suited to deal with social network data, since Wikipedia pages and social network posts generally do not share the same writing style nor the same vocabulary. Yet, the lack of annotated social network corpora, especially in French, will inevitably force us to rely on the Wikipedia dataset, since it is the only publicly available corpus with direct mappings between entities and mentions spans. Nevertheless, Wikipedia is insufficient on its own. As pointed out by [57], better performances are obtained with a combination of the Wikipedia dataset and a small social network dataset, `WNUT2017` [58]. We took these findings into account and we build a *natural risk aware entity-linking dataset* by merging the following datasets:

- The French Wikipedia corpus

- The HIPE-2022 dataset

- The CAp2017 dataset

- Our own annotated Twitter dataset extracted during a major French natural crisis

## 4.1 The French Wikipedia dataset

The Wikimedia Foundation regularly publishes dumps of the current state of their encyclopedia (https://dumps.wikimedia.org/). Until recently, Wikipedia articles were available only as wikitext, the markup language in which Wikipedia pages are written. This language is known to have no clearly defined syntax and extensively relies on (nested) template expansions, making it extremely difficult to write a reliable parser. Indeed, previous work shows that standard parsing methods could not reproduce the expected Wikipedia output [59].

Fortunately, this era has ended since the release of the Wikipedia Enterprise HTML dumps in 2021, providing easy access to the whole Wikipedia corpus as it is available online, without the need for a tedious an unreliable parsing step.

Even though parsing wikitext is no longer required, pages must be processed in order to remove HTML markups and do some light cleaning. We removed all headings, figures, tables and equations as well as some *noisy* sections: bibliography, references, see also, appendix and external links. Hyperlinks targeting another Wikipedia page have been considered as mention span annotations and the target has been used as the target entity. We also extracted the first sentence of each page to produce a dictionary-like dataset mapping entities to their descriptions, as well as other information such as their Wikipedia and Wikidata identifiers. Finally, we kept only the first section of each Wikipedia page, in order to keep the dataset's size reasonable.

While such a dataset would probably be sufficient to train a generic Entity-Linking system, it does not fulfil all of our application requirements. First, we need a mapping between Wikipedia titles and Wikidata identifiers, in order to be able to extract GPS coordinates from the Wikidata knowledge base. Second, as is, the dataset does not give any clue on the type of annotated mentions, such as the ones used in NER. Indeed, we need to know whether a mention refers to a location, a facility, or a natural hazard or if it describes the effects of a natural hazard on goods or people.

**Mapping between Wikipedia and Wikidata.** Many data related to Wikipedia pages are stored in a relational database. In addition to the Wikipedia dumps mentioned beforehand, the Wikimedia Foundation publishes a set of SQL dumps from which one can load all the data stored in their databases. In particular, they store page properties in an SQL table called *page_props*, page redirects in a table called *redirect* and, finally, pages are stored in the *page* table. Extensive documentation of the database layout is available on the MediaWiki wiki (https://www.mediawiki.org/wiki/Manual:Database_layout).

Many Wikipedia pages are redirect pages. From the user's point of view, redirects are almost invisible and a page and any of its redirects are interchangeable. For instance, the page *Leucetia* automatically redirects to *Paris*. Redirect pages can also redirect to other redirect pages, but these pages are not associated with any Wikidata entities, only the *real* pages are. Hence, computing all the redirect paths is required in order to link mentions to their real target Wikipedia pages and, consequently, get the target Wikidata identifiers.

Redirect paths can be computed from the *redirect* table. Then, Wikidata identifiers can be retrieved from the *page_props* table. We also retrieve additional information, such as Wikipedia identifiers and page names from the *page* table.

**Annotating mentions with types.** We considered a set of nine NER labels that focus on the types of information sought in tweets during natural disasters, following the recommendations of real practitioners. Where possible, NER labels were defined with enough overlap with standard NER labels such that existing datasets can be used to train our model with minimal effort as in the case of geographic information. The other labels correspond to entities specific to natural disaster management, that are well represented in the tweets. Plus, even though

`PERSON` and `ORG` labels might seem unnecessary at first sight, they do convey geographical knowledge since they may refer to local representatives (e.g. the mayor of a municipality or associations established in a well-known area).

**PERSON** real or fictive characters

**ORG** organization, such that press organisms, companies or association

**GEOLOC** geographic entities

**TRANSPORT** lines such as roads, railways or rivers

**EVENT** events such as a festival

**FACILITY** facilities such as a nuclear power plant

**RISKNAT** mentions identifying natural hazards, such as their manifestations (wind, rain, flooding, ground motion, etc.), their intensity and magnitude, or their forecast (early warnings)

**DAMAGES** effects of natural hazards on goods (destruction, damage, malfunction, closure, etc.) or people victims, missing persons, etc.)

**OTHER** mentions that do not fall in any of the previous categories

Since links in Wikipedia pages do not have any sort of label attached to them, it is not possible to train a NER model on the raw Wikipedia corpus. We decided not to rely on off-the-shelf NER models, since they were not trained to detect some kind of entity mentions, especially RISKNAT and DAMAGES, which are crucial for our end application. Furthermore, mentions of the same entity used in different contexts could be given different labels since NER models are not necessarily consistent, even if they are generally very effective. Instead, we manually annotated few Wikipedia pages and fine-tuned a pre-trained CamemBERT model to classify a Wikipedia entity according to the description given by the first sentence of its Wikipedia page. We found out that annotating no more than a thousand entities was sufficient. More precisely, 1192 pages were manually annotated. We first build a test set with 10% of the annotated data and found out that the model trained on the remaining 90% achieved around 0.75 Fscore (precision was 0.8 and recall 0.76). We then trained a classifier on the whole dataset and applied it to classify all the French Wikipedia entities. Finally, we updated our EL dataset accordingly. A summary of the mentions present in the corpus is given in Table 1.

**Table 1. An overview of the mentions annotated in the Wikipedia dataset.** *#Mentions* shows the total number of mentions per label, *#Linked* the number of mentions linked to an entity and *#Entities* the number of distinct entities per label present in the dataset.

| Labels | #Mentions | #Linked | #Entities |
|---|---|---|---|
| PERSON | 1098143 | 1098143 | 557561 |
| ORG | 748714 | 748714 | 130113 |
| GEOLOC | 2724062 | 2724062 | 215223 |
| TRANSPORT | 160284 | 160284 | 53304 |
| EVENT | 798230 | 798230 | 86455 |
| FACILITY | 258419 | 258419 | 109818 |
| RISKNAT | 0 | 0 | 0 |
| DAMAGES | 0 | 0 | 0 |
| OTHER | 4339458 | 4339458 | 682412 |
| Total | 10127310 | 10127310 | 1834886 |

While the data offered by Wikipedia is extremely precious, it is actually not sufficient for many use cases, especially when the target application consists of dealing with specific or technical documents. In our case, training a model on Wikipedia does not allow the model to understand tweets, since messages posted on Twitter do not share the same structure as Wikipedia sentences nor use the same vocabulary. Furthermore, tweets are often full of misspelling errors and can be redacted using slang unseen on Wikipedia. As such, the Wikipedia corpus can be used as a strong foundation to train generic models but is not sufficient to deal with more specific content. As such, we considered extending the Wikipedia corpus with more specific corpora.

## 4.2 The HIPE-2022 dataset

The HIPE-2022 task aims at identifying entities from historical documents. As part of the challenge, a multilingual corpus with mention annotations has been released. The corpus has been created by merging six annotated datasets: AjMC [60], CLEF-HIPE-2020 [61], LeTemps [62], TopRes19th [63], NewsEye [64] and Sonar [65]. As our work focuses specifically on French, we consider only the French part of the whole dataset, hence discarding TopRes19th and Sonar as well as the non-French documents of the four others. Mentions are annotated both for the NER and the EL tasks, making the dataset particularly suitable to train our dual-task model. An overview of the mentions annotated in the dataset is given in Table 2.

Some adjustments to the dataset have been required because the HIPE-2022 NER labels do not match ours. Thus, we mapped the HIPE-2022 labels to the ones presented earlier in the paper. The mapping is given in Supporting Information.

While the dataset focus is absolutely not relevant to our target application, we do think that diversifying the training data should help the model to better handle real-world data. Furthermore, considering the HIPE-2022 dataset allows us to compare our model with the models submitted as part of the HIPE-2022 challenge.

## 4.3 The CAp 2017 NER dataset

The CAp 2017 challenge concerns the problem of Named Entity Recognition for tweets written in French [66]. An overview of the released dataset is shown in Table 3.

Labels used in this dataset are relatively fine-grained and not necessarily relevant to our use case, hence we replaced them with the same set of labels defined in the previous section. The complete mapping is given in Supporting Information.

**Table 2. An overview of the mentions annotated in the French subset of the HIPE-2022 dataset.** *#Mentions* shows the total number of mentions per label, *#Linked* the number of mentions linked to an entity and *#Entities* the number of distinct entities per label present in the dataset.

| Labels | #Mentions | #Linked | #Entities |
|---|---|---|---|
| PERSON | 14325 | 5096 | 1678 |
| ORG | 3922 | 1845 | 510 |
| GEOLOC | 15020 | 8540 | 1658 |
| TRANSPORT | 195 | 68 | 30 |
| EVENT | 0 | 0 | 0 |
| FACILITY | 293 | 80 | 29 |
| RISKNAT | 0 | 0 | 0 |
| DAMAGES | 0 | 0 | 0 |
| OTHER | 2918 | 870 | 118 |
| Total | 36673 | 16499 | 4023 |

**Table 3. An overview of the mentions annotated in the CAp 2017 dataset.** *#Mentions* shows the total number of mentions per label.

| Labels | #Mentions |
|---|---:|
| PERSON | 1644 |
| ORG | 1348 |
| GEOLOC | 1265 |
| TRANSPORT | 1005 |
| EVENT | 168 |
| FACILITY | 287 |
| RISKNAT | 0 |
| DAMAGES | 0 |
| OTHER | 940 |
| Total | 6657 |

Since the challenge focuses on the NER task, mentions were not linked to their corresponding Wikipedia entities. Hence, it will not be helpful with regard to the EL task. Nonetheless, our proposed model aims at both detecting and classifying mention (NER) and linking them to a target knowledge base (EL), hence it could benefit from being trained on the CAp2017 dataset. Indeed, the dataset has been built such that it contains a broad diversity of tweets, it then should be extremely valuable and should help the model build coherent representations for tweets, as well as detecting and classifying mention spans in tweets, which is the first step to a good EL model. Yet, the lack of crisis-related information and EL annotations will bound us to build our own dataset.

## 4.4 The RéSoCIO dataset

The three datasets presented in the previous sections are not particularly adapted to our target application: geolocate tweet content when related to a natural disaster. First, social network posts are largely underrepresented in these datasets, and the only Twitter dataset lacks Entity-Linking annotations. Second, none of the datasets focuses on a crisis or natural disaster event.

To mitigate these issues, we extracted a collection of French tweets written during earthquakes and major floods that have occurred in France in recent years. We set up Label-Studio [67] in order to annotate these tweets. A total of 4617 tweets were annotated, including 1678 tweets posted during earthquakes and 2939 during floods. For each annotated tweet, mentions were annotated using the set of labels described earlier in the paper as well as, when possible, the target Wikipedia title.

Named *"RéSoCIO"* in reference to the research project in which it was carried out, the dataset resulting from this work contains a total of 12828 annotated mentions and 1513 distinct Wikipedia entities. The data collection and analysis method used to compile this dataset—which is available on the ZENODO platform (https://doi.org/10.5281/zenodo.7767294)—complies with the data usage terms and conditions defined for the Academic Research access to the Twitter API.

In this dataset, 85% of mentions were associated with a Wikipedia page and 94% if we ignore the RISKNAT and DAMAGES labels, which are often difficult to map to an existing entity. The precise breakdown of the number of mentions per label is given in Table 4.

Finally, we build our training dataset by concatenating all the datasets presented in this section. An overview of this dataset is given in Table 5.

**Table 4. An overview of the mentions annotated in the Twitter dataset.** *#Mentions* shows the total number of mentions per label, *#Linked* the number of mentions linked to an entity and *#Entities* the number of distinct entities per label present in the dataset.

| Labels | #Mentions | #Linked | #Entities |
|---|---|---|---|
| PERSON | 315 | 263 | 136 |
| ORG | 863 | 790 | 281 |
| GEOLOC | 4375 | 4234 | 701 |
| TRANSPORT | 250 | 203 | 101 |
| EVENT | 35 | 21 | 16 |
| FACILITY | 129 | 94 | 49 |
| RISKNAT | 5502 | 4994 | 128 |
| DAMAGES | 1136 | 121 | 56 |
| OTHER | 223 | 200 | 46 |
| Total | 12828 | 1322 | 1513 |

# 5 Entity-linking pipeline evaluation

## 5.1 Experimental setup

The bi-encoder architecture has to detect mentions of interest in tweets and associate them with the right corresponding Wikidata entity. To evaluate the model's performance on these two tasks, an additional set of 339 tweets was manually annotated considering the same instructions used for labelling the RéSoCIO dataset. Half of these tweets were written during a storm affecting Corsica, and the other half during The Teil earthquake. Table 6 shows an overview of these data, restricted to the most impactful classes of this study.

Both mention and entity encoders have been preloaded with CamemBERT's pre-trained weights [68]. However, as proposed by [15], we only loaded the first four layers, instead of all the twelve layers, making our model smaller while being finetuned to solve multiple tasks at once. The model has been trained using the AdamW optimizer with a batch size of 32 on a single NVidia Tesla V100 GPU. The model may benefit from being trained with larger batch size as well as with hard-negatives mining [69].

The Entity-Encoder (in the bi-encoder model) has been trained with entity descriptions from Wikipedia. We assumed that the Wikipedia guideline stating that the first sentence of an article should be a short description (https://en.wikipedia.org/wiki/Wikipedia:Manual_of_Style/Lead_section) of the topic is followed by most articles. We thus consider only the first

**Table 5. An overview of the mentions annotated in the full dataset.** *#Mentions* shows the total number of mentions per label, *#Linked* the number of mentions linked to an entity and *#Entities* the number of distinct entities per label present in the dataset.

| Labels | #Mentions | #Linked | #Entities |
|---|---|---|---|
| PERSON | 1100102 | 1098406 | 557697 |
| ORG | 750925 | 749504 | 130394 |
| GEOLOC | 2729702 | 2728296 | 215924 |
| TRANSPORT | 161539 | 160487 | 53405 |
| EVENT | 798433 | 798251 | 86471 |
| FACILITY | 258835 | 258513 | 109867 |
| RISKNAT | 5502 | 4994 | 127 |
| DAMAGES | 1136 | 121 | 56 |
| OTHER | 4340621 | 4339658 | 682458 |
| Total | 10146795 | 10138230 | 1836399 |

**Table 6. An overview of the mentions annotated in the test dataset.** *#Mentions* shows the total number of mentions per label, *#Linked* the number of mentions linked to an entity and *#Entities* the number of distinct entities per label present in the dataset.

| Labels | #Mentions | #Linked | #Entities |
|---|---|---|---|
| GEOLOC | 384 | 365 | 107 |
| RISKNAT | 477 | 380 | 29 |
| DAMAGES | 124 | 8 | 6 |
| Total | 985 | 753 | 142 |

sentence to describe entities instead of the ten first as proposed by [43]. For instance, the entity `Paris` would be described by the following sentence: "Paris is the capital and most populous city of France, with an estimated population of 2165423 residents in 2019 in an area of more than 105 km$^2$, making it the 30$^{th}$ most densely populated city in the world in 2020.".

The cross-encoder has been trained on negative examples, or erroneous candidates, generated by the bi-encoder. We built a training dataset of (*mention*, *entity*) pairs by predicting candidates entities of an annotated mentions. For each annotated mention, we retrieved the top 10 candidates and kept the wrong ones as negative examples. We still included the positive pair even if the bi-encoder was not able to retrieve the target entity. This kind of models are usually trained using contrastive learning methods, by relying on the other samples in the batch to generate negative examples. We found out that training our cross-encoder this way was particularly ineffective, since the training setup deviate too much from the real application setup. Indeed, the cross-encoder should be able to select the right entity among a set of very similar candidates generated by the bi-encoder. Yet, the entities sharing the same batch are, most of the time, very different, making the training task too easy and actually extremely different from the target task. As a consequence, the cross-encoder trained on negative samples generated by the bi-encoder was a lot better than the one trained using contrastive learning.

## 5.2 Results

When the strong imbalance observed between the different categories of entities is taken into account, performances on the NER task are good with an F1 score at 0.83 (see micro scoring in Table 7). Although there is no benchmark on French tweets on which to base the results of this work, it is possible to consider state-of-the-art performances on NER on English tweets to position the performance of the Bi-Encoder for the NER task. Indeed, a cross-language study on Wikipedia showed that NER results do not vary so much between French and English [70]. Recently, a comparison of several NER models showed that state-of-the-art F1 scores are smaller than 75% for tweets [71], meaning that the results of the Bi-Encoder are excellent. The scores by class show contrasted predicting performances between the different types of entities (Table 8. Unsurprisingly, the RISKNAT and GEOLOC classes, which are the most represented in the training set, are the best predicted, with F1 scores of 0.82 and 0.70 respectively. The low recall observed for the DAMAGES class may be related to the complexity of the mentions in

**Table 7. Micro and macro scores of our model on the NER task and performances on the EL task.** R@1, R@5, R@10 and R@100 indicate the proportion of relevant predictions in the 1, 5, 10 and 100 candidate entities whose representations are the closest to the one of the related mention.

| Metrics | NER | | | Entity linking | | | |
|---|---|---|---|---|---|---|---|
| | Precision | Recall | Fscore | R@1 | R@5 | R@10 | R@100 |
| Micro | 0.83 | 0.83 | 0.83 | 0.63 | 0.81 | 0.84 | 0.84 |
| Macro | 0.63 | 0.39 | 0.56 | | | | |

**Table 8. Per-class performances of the Bi-encoder model on the NER and EL tasks.** R@1, R@5, R@10 and R@100 indicate the proportion of relevant predictions in the 1, 5, 10 and 100 candidate entities whose representations are the closest to the one of the related mention.

| Labels | NER | | | Entity-linking | | | |
|---|---|---|---|---|---|---|---|
| | Precision | Recall | Fscore | R@1 | R@5 | R@10 | R@100 |
| GEOLOC | 0.80 | 0.63 | 0.70 | 0.66 | 0.85 | 0.88 | 0.88 |
| RISKNAT | 0.87 | 0.77 | 0.82 | 0.66 | 0.84 | 0.86 | 0.86 |
| DAMAGES | 0.86 | 0.29 | 0.43 | 0 | 0 | 0 | 0 |

this class, whose structure can vary greatly (words, number of words, etc.) making them potentially difficult to identify for the model. The relatively low recall score for GEOLOC can be explained by the complexity of tweets where GEOLOC mentions are often abbreviated or nicknamed. For instance, in our tweets, "stras" is a GEOLOC mention that refers to the city of Strasbourg, and this mention is not detected by our model.

Performances on the EL task were evaluated using the metric Recall@$k$ (R@$k$) which usually indicates the proportion of relevant predictions in the first $k$ results. In the EL task, for any mention $m$, there is at most one relevant entity among a set of $k$ candidate entities. These candidate entities correspond to $k$ entities whose representations, computed by the Entity Decoder, are the closest to the representation of $m$ produced by the Mention Encoder. In these conditions, a R@100 value of 0.81 indicates that in 81% of cases, the entity to be predicted appears in the first 100 candidates returned by the model. Worth noting that the representations for the entities are calculated beforehand, by providing textual descriptions to the Entity Encoder, once it has been once it has been trained. The Entity Encoder is thus not used to link an entry to its entity. However, it remains indispensable to calculate the representations of any new entities.

The evaluation of the performance of the EL task is less well standardised than that for the NER task. Thus, while it is possible to give an order of magnitude of the state of the art on the EL task based on the EL work carried out on English tweets and that carried out on the recognition of spatial entities, metrics in those papers may differ somehow from the ones used in this work. For example, Hebert and his collaborators assessed recently the robustness of several approaches on an entity linking benchmark constituted of English tweets [72]. For an approach similar to our Bi-Encoder, they presented an R@16 of 0.759, which was level up to 0.887, by hybridising the model with a sparse retrieval strategy. Performances obtained on French tweets with the Bi-Encoder, *i.e.* R@10 of 0.84 are thus very competitive (Table 7).

R@*1* scores show that more than two-thirds of mentions are associated with the right entity. It is interesting to note that the entity to be found is, in 81% of cases, one of the first five candidate entities, which suggests that in case of misleading location/results, the exploration of the first five candidate entities returned by the model may be sufficient to find the right result in an applicative context.

Finally, regarding the by-class performances (Table 8), the more representative results on the EL task are supported by the GEOLOC label. Indeed, almost all mentions related to DAMAGES cannot be linked to any Wikipedia entity because of their nature, which explains the results observed for this label. The case of the RISKNAT label is a little bit different because mentions related to this label correspond to information describing the phenomenon (e.g. "heavy rain", "strong gusts of wind", "magnitude of 4.5" etc.) which could be linked to very general French Wikipedia entities such as "Rain", "Thunderstorm" and so on, explaining goods results on this label. A recent paper assessed the performances of several BERT-based language models used for geo-entity linking [73], which aim to link mentions similar to the

**Table 9. Total number of tweets and number of tweets, mentions and entities predicted by our model.** The number between parenthesis indicates the number of tweets, mentions or entities which have been localized inside the area impacted by the earthquake/storm.

| Event | #Tweets | Geolocalized | | |
| --- | --- | --- | --- | --- |
| | | Tweets | Mentions | Distinct entity |
| Teil | 17599 | 10673 (4791) | 15341 (6732) | 895 (99) |
| Alex | 15874 | 9349 (2961) | 14643 (3618) | 1223 (150) |

ones associated with the GEOLOC label. Authors of this work reported R@1, R@5 and R@10 scores of 0.402, 0.744 and 0.864 respectively, showing the results of the Bi-Encoder on the EL task are systematically better.

# 6 Geo-location of the effects of natural disasters

To evaluate the behaviour of our model in real-world situations, we selected two major natural disasters that hit France very recently: namely the Le Teil earthquake and storm Alex. We extracted for both crises a set of tweets emitted throughout a period of one month centred on each event. The tweets were collected using the Twitter Academic API using the query given in Supporting Information. An overview of the collected tweets is given in Table 9.

Since no ground truth is available to quantitatively assess the performances of our model, we prefer to propose, in this section, a qualitative analysis of the geolocation predictions made by the model. We will mostly verify that the predictions match an expected behaviour, which can be summarised by the following pattern:

**Before events**: Low number of tweets using the lexical fields of earthquakes or floods/storms, and high geographical dispersion of toponyms mentioned in these messages.

**During / right after events:** Spike in Twitter activity and concentration of detected toponyms around the impacted areas.

**After events:** Decrease in the number of tweets and increase in the geographical dispersion of detected toponyms, towards the restoration of the initial state.

In other words, in the absence of natural events, we expect Twitter activity to remain calm and noisy, and conversely, when an event occurs, we suppose that the Twitter activity will suddenly peak and focus on the impacted area.

In the following, we will consider only the predictions made by Model C, described in the previous section. Our model predicts mention spans, with labels, as well as target entities, but not GPS coordinates. However, coordinates can be easily drawn from Wikidata entities by extracting the `P625` Wikidata property (https://www.wikidata.org/wiki/Property:P625), namely the "coordinate location" property. Since our model detects non-geographical entities too, we filtered its predictions to keep only the mentions classified as one of GEOLOC.

## 6.1 Teil earthquake

**6.1.1 Event presentation.** With a magnitude Mw of 4.9 according to the European-Mediterranean Seismological Centre (EMSC), the Teil earthquake on the 11[th] November of 2019 is the most powerful earthquake having occurred in mainland France for over 20 years, and the most destructive for 50 years. Located in the east of the Ardèche department, this earthquake, which generated very significant damages at the epicentre, was felt in a large part of south-eastern France.

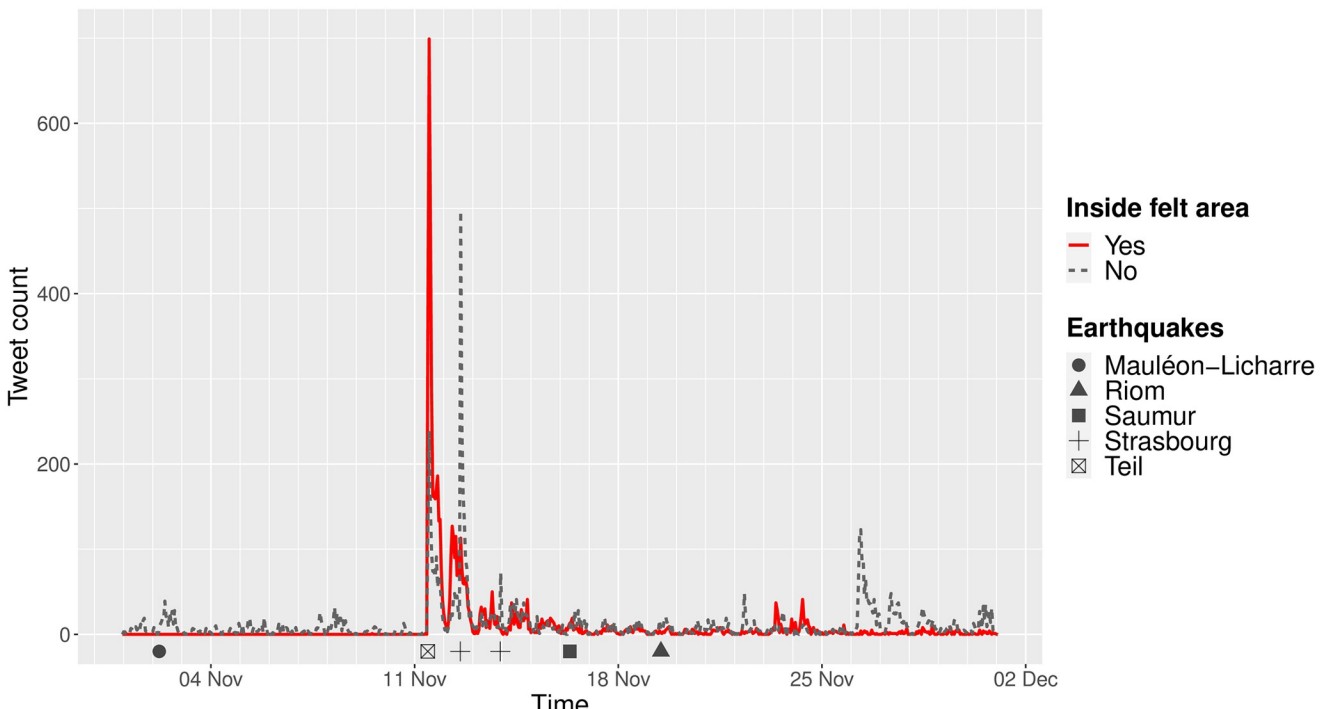

**Fig 3. Geolocalized mentions count over time in November 2019.** The plain red line represents mentions localized in the Teil earthquake felt area. Earthquakes seem to trigger peaks of Twitter activity.

**6.1.2 Analysis of the geographical signature from the tweets.** We first examined the evolution over time of the frequency of tweets mentioning keywords from the French lexical field related to earthquakes, before, during and after the Le Teil earthquake (from November 1st to 30th). For each of the five earthquakes that occurred during the study period (on November 2nd, 11th, 12th, 13th and 16th), we delineated the areas within which the seismic shaking was felt by the population. To do this, we analysed the data provided by the French Central Seismological Office (BCSF), linking together localities with a macro-seismic intensity value greater than or equal to 3 according to the EMS-98 scale [74]. In particular, this allows us to discriminate the spatial entities identified in the tweets according to whether or not they are located in the perception zone of the Le Teil earthquake. As clearly shown in Fig 3, a massive spike in Twitter activity occurred immediately after the Le Teil earthquake, essentially within the perception zone of the earthquake: a detailed examination shows that this activity peak is recorded at 10:55, only 3 minutes after the occurrence of the earthquake at 10:52. Beyond the case of the Le Teil earthquake, we also observe a non-negligible increase in the frequency of tweets after each of the four earthquakes that hit France during the period studied (Fig 3). This observation tends to validate the hypothesis that monitoring the frequency of tweets alone could be almost sufficient to detect such fast kinetic events [75], especially in densely populated areas such as Lyon (where the shaking from the Le Teil earthquake was weakly felt and reported on Twitter) and Strasbourg (where two small earthquakes induced by geothermal activity were felt successively on November 12th and 13th 2019). In addition to these peaks of activity, it is also interesting to note in Fig 3 a peak of activity on November 26th, when no earthquake occurred in France: these tweets signal the occurrence of a powerful earthquake which caused considerable damage in Albania.

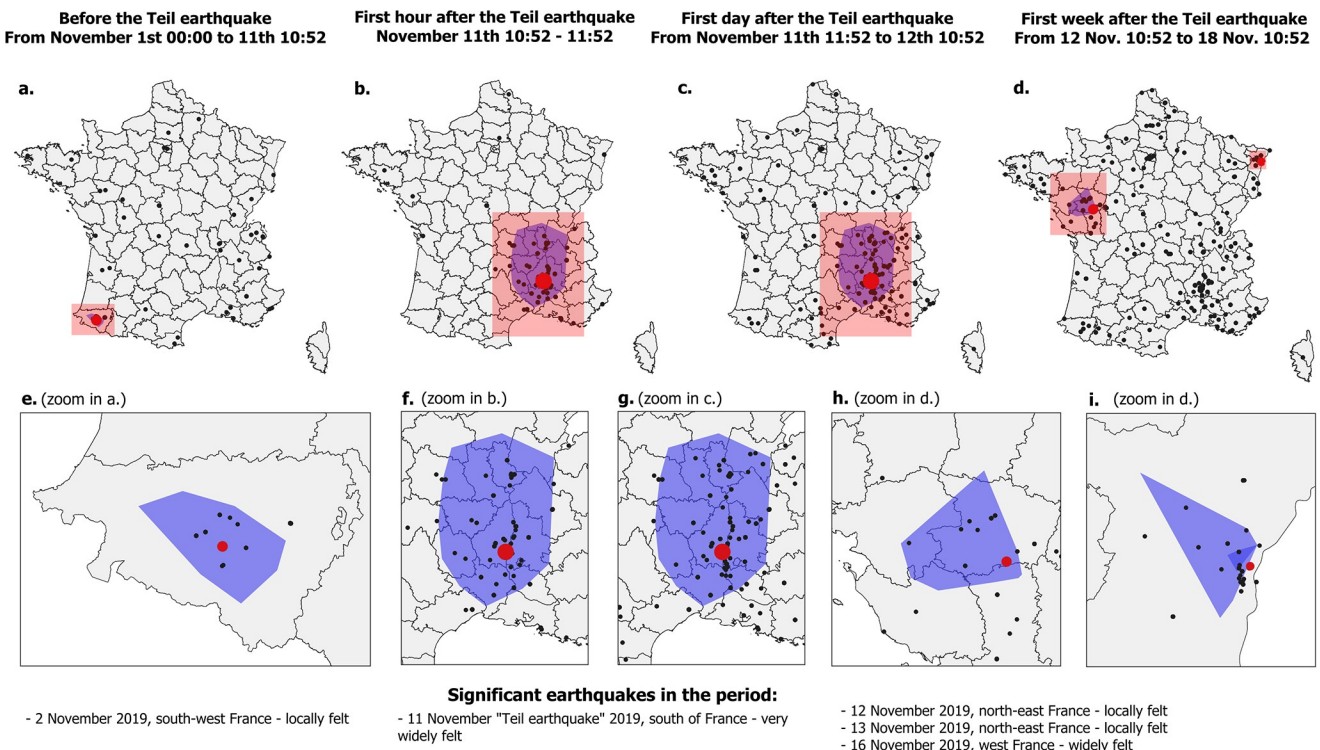

**Before the Teil earthquake**
**From November 1st 00:00 to 11th 10:52**

**First hour after the Teil earthquake**
**November 11th 10:52 - 11:52**

**First day after the Teil earthquake**
**From November 11th 11:52 to 12th 10:52**

**First week after the Teil earthquake**
**From 12 Nov. 10:52 to 18 Nov. 10:52**

a.

b.

c.

d.

**e.** (zoom in a.)

**f.** (zoom in b.)

**g.** (zoom in c.)

**h.** (zoom in d.)

**i.** (zoom in d.)

- 2 November 2019, south-west France - locally felt

**Significant earthquakes in the period:**
- 11 November "Teil earthquake" 2019, south of France - very widely felt

- 12 November 2019, north-east France - locally felt
- 13 November 2019, north-east France - locally felt
- 16 November 2019, west France - widely felt

**Fig 4. Geo-localized mentions detected in tweets issued one month before the Teil earthquake (a,e), then in the first hour (b,f)/ day (c,g)/ week (d,h) after the earthquake.** Also shown are the epicentres of significant earthquakes that occurred during each period (red circles proportional to the magnitude of each earthquake) as well as the areas in which the ground motions were felt by the population for each earthquake (blue polygons), estimated from BCSF data. Outline of French departments from OpenStreetMap contributors under ODbL license.

Hence, according to our model, a large majority of tweets issued immediately after the Le Teil earthquake mentioned locations in the perception area, while almost none referred to such locations before. This suggests that our model's predictions are coherent with reality since the Teil and the other perception areas are mentioned only after the earthquakes happened and during a relatively short time.

In order to validate this second observation, we first represent the map of place names detected by our model within the tweets written before the Teil earthquake. Maps in Fig 4a–4e show that tweets seem to be located relatively randomly with a slight concentration near Paris, the French capital and the most populated city, and in the south-west, where the Mauléon-Licharre earthquake happened on November 2$^{nd}$. In contrast to this "quiet" period, during which Twitter activity highlights widely dispersed localities, **a massive concentration of tweets localized near the felt area can be observed in less than 10 minutes after the earthquake has been felt**, as shown in Fig 4b–4f. In agreement with Fayjaloun et al. [13] who suggest that the information exchanged on Twitter in the first ten minutes after an earthquake is most often sufficient to characterise its spatial footprint, half of the impacted area is covered by our model predictions after 10 minutes and almost the entire is covered after only one day (in fact, this is already the case after 1 hour). After one day, the locations predicted are more sparse and random, while forming clear clusters near areas impacted by other earthquakes, such as Strasbourg (Fig 4d–4h) and Saumur (Fig 4d–4i).

While this last analysis does not prove that our model predictions are correct, it does show that they are at least coherent with what really happened, especially considering that very few

predictions are localized outside the felt area once the Teil earthquake happens. Indeed, less than 10% of the model's predictions on tweets written at most 10 minutes after the earthquake is outside the impacted area. The ratio of tweets outside the impacted area increases to 15% when considering the tweets written between 10 and 60 minutes after the earthquake.

## 6.2 Alex storm

**6.2.1 Event presentation.** The Alex storm hit France from Brittany on 1$^{st}$ to 2$^{nd}$ October 2020. Then it moved southwards on 2$^{nd}$ October, bringing intense and stormy rainfall until 3$^{rd}$ October, mainly in the Alpes-Maritimes department. This storm caused flash flooding, resulting in exceptional damage and numerous victims in the high valleys above the city of Nice. Storm Alex also affected other departments in the east and southwest of France, as well as Italy and Switzerland.

**6.2.2 Analysis of the geographical signature from the tweets.** As for the Teil earthquake, we analysed the frequency of tweets mentioning keywords from the French lexical field related to floods (i.e. the most devastating effect from the storm), as shown in Supporting Information, over a one-month window centred on the Alex storm. During this time span, major floods happened in the Gard department between 19 and 20 September 2020. In a similar way to the seismic phenomenon for which we delimited the perception area of shaking from macroseismic intensity data, for the floods, we instead considered the areas with high rainfall accumulation, as calculated by Météo France. Thus, areas, where a cumulative rainfall of more than 80 mm was recorded in 72 hours between 2 and 5 October, were considered to be impacted by the Alex storm.

As can be seen in Fig 5, spikes in Twitter activity occurred after the Alex storm and the floods in Gard. In particular, we observe a notable increase of tweets referring to toponyms

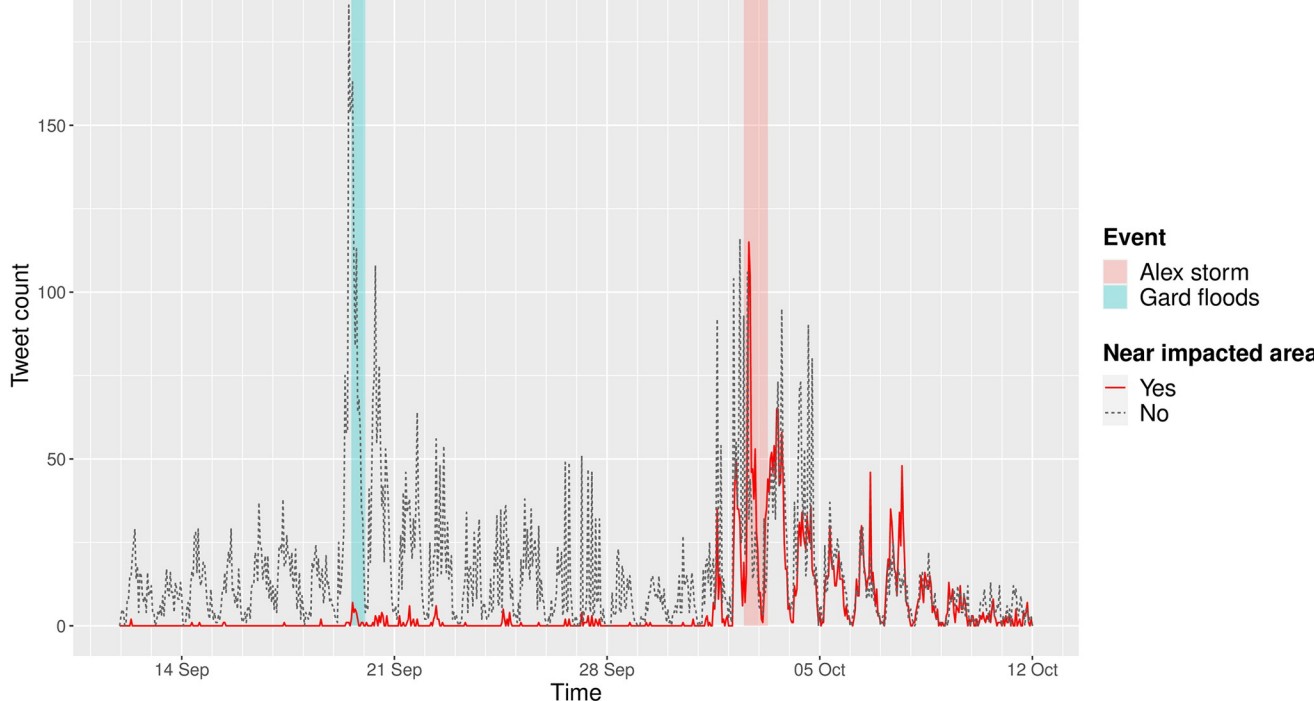

**Fig 5. Geolocalized mentions count over time from September to October 2019.** The plain red line represents mentions localized near areas impacted by the Alex storm.

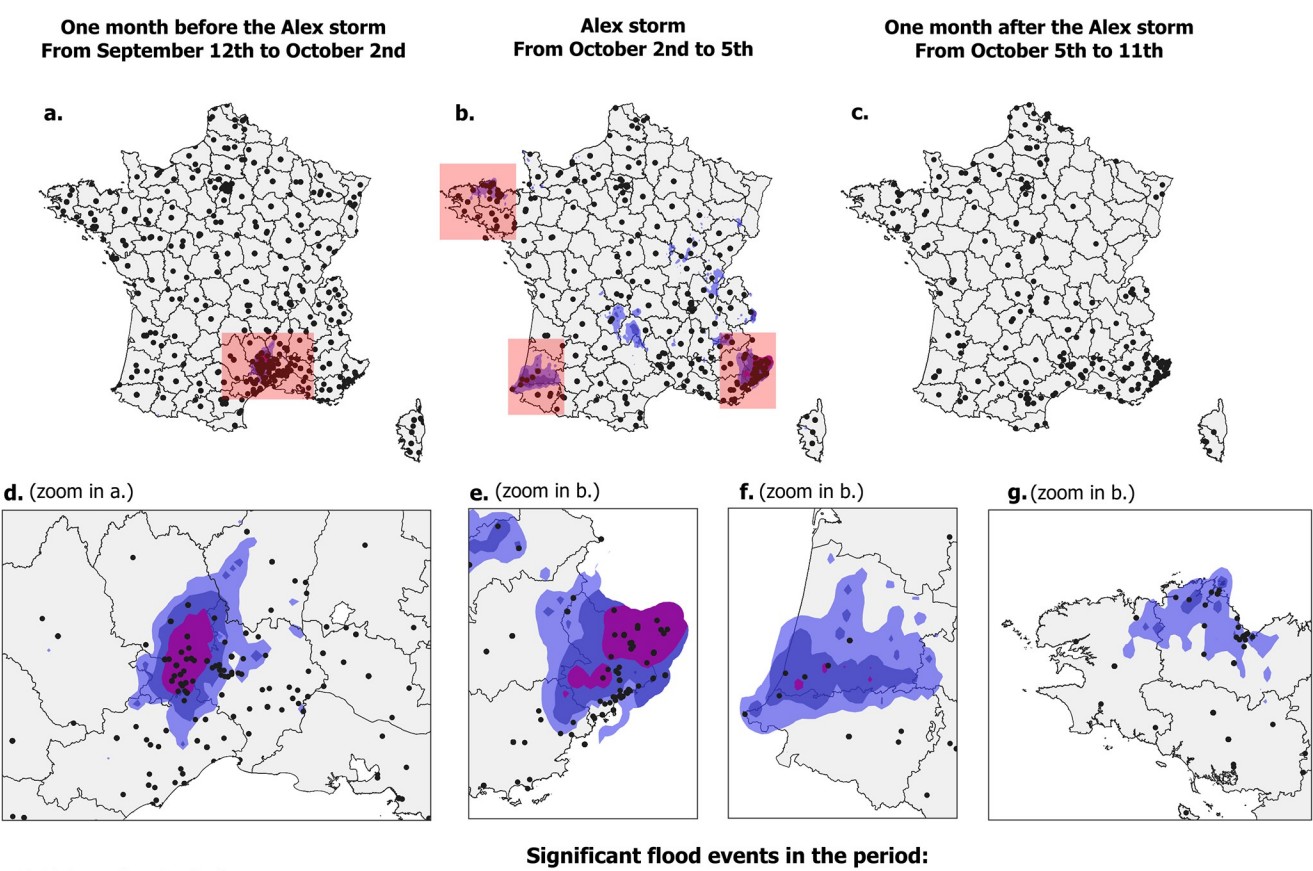

**Fig 6. Geo-localized mentions detected in tweets issued one month before the Alex storm (a,d), then during (b,e,f,g) and after (c) the storm.** Also shown are the areas with the highest rainfall recorded by Météo France during the peak of the Alex storm (cumulative rainfall from 2 October 6h to 5 October 6h) and during a previous episode of heavy rains that led to flooding in the South of France (cumulative rainfall from 18 September 6h to 21 September 6h): over 80 mm—light blue polygons, over 100 mm—dark blue polygons, over 160 mm—purple polygons. Outline of French departments from OpenStreetMap contributors under ODbL license.

geolocated near the area impacted by the Alex storm after it hit France, while those locations were almost never mentioned during the previous event on September 19[th], which match, once again, our expectations.

However, the behaviour of Twitter users seems to differ from what we observed previously. Indeed, spikes occur as soon as an earthquake is felt by the population, and the activity fades quickly. This is probably due to the sudden and unpredictable nature of earthquakes. Conversely, while properly speaking weather conditions cannot be truly predicted, they can be forecasted. As a result, the Alex storm was discussed on Twitter before it actually hit the territory, and the Twitter activity reached its peak when the Alex storm did actually happen, as shown in Fig 5.

The analysis of the corresponding maps shows first of all in Fig 6a that, as in the case of the earthquakes, the activity is very dispersed before the event occurs, but with a higher level of "noise". Moreover, the flooding episode of 19 and 20 September 2020 in the Gard department is reflected in this background noise by a greater geographical concentration of toponyms used in tweets in the south of France (Fig 6a–6d). Between October 2[nd] and 5[th], at the height of the passage of storm Alex in France, and while a high level of noise remains in the geolocation

data extracted by our models, geographical concentrations appear in the different areas most affected, mainly in the south-east (Fig 6b–6e), the south-west (Fig 6b–6f) and the north of Brittany (Fig 6b–6g). After 5 October, this geographical concentration diminished, although there was still marked activity in the southeast (Fig 6c), while a wave of solidarity for the victims was felt throughout France.

## 6.3 Discussion

Our model seems to be able to capture coherent representations of real natural disasters. Hence, in the absence of an event, Twitter activity remains relatively flat and the few location predictions are randomly located, albeit with an over-representation of densely populated areas. In the event of a fast kinetic event such as a rapid flood or earthquake, activity on Twitter increases dramatically, with the focus of messages on the affected areas. This behaviour is clearly well characterised by our model.

Table 10 provides a quantitative illustration of this trend. It shows a significant increase in the spatial density of mentions in areas affected by earthquakes or heavy rain. In the case of

**Table 10. Comparison, for the earthquake and heavy rainfall events considered in figure #, between the density of geo-localized mentions inside and outside the affected areas.** For earthquakes, the analyses were made considering all tweets captured during the calendar day of the earthquake (i.e. November 11, 2019 for the Teil earthquake). For episodes of heavy rainfall, tweet exports correspond strictly to the time windows of cumulative rainfall.

| Event | Period | Area | Mentions | Area (km2) | Density (mentions/ km2) | Ratio relative to background noise in unaffected area |
|---|---|---|---|---|---|---|
| Earthquake | 2 Nov. | Unaffected area | 54 | 547133 | 9.9E-05 | 1.0 |
| | | Felt area | 45 | 1728 | 2.6E-02 | 263.9 |
| | 11 Nov. | Unaffected area | 840 | 498689 | 1.7E-03 | 1.0 |
| | | Felt area (Teil EQ) | 2991 | 50171 | 6.0E-02 | 35.4 |
| | 12 Nov. | Unaffected area | 520 | 498234 | 1.0E-03 | 1.0 |
| | | Felt area—Teil EQ | 1396 | 50171 | 2.8E-02 | 26.7 |
| | | Felt area | 825 | 456 | 1.8E+00 | 1734.7 |
| | 13 Nov. | Unaffected area | 284 | 498668 | 5.7E-04 | 1.0 |
| | | Felt area—Teil EQ | 356 | 50171 | 7.1E-03 | 12.5 |
| | | Felt area | 4 | 22 | 1.8E-01 | 323.5 |
| | 16 Nov. | Unaffected area | 119 | 541910 | 2.2E-04 | 1.0 |
| | | Felt area—Teil EQ | 133 | 50171 | 2.7E-03 | 12.1 |
| | | Felt area | 23 | 6950 | 3.3E-03 | 15.1 |
| Heavy rains | From 2020–09-18 06:00 to 2020–09-21 06:00 | Unaffected area | 1651 | 543320 | 3.0E-03 | 1.0 |
| | | Cum. rainfall ≥ 80 | 9 | 2689 | 3.3E-03 | 1.1 |
| | | Cum. rainfall ≥ 100 | 399 | 1672 | 2.4E-01 | 78.5 |
| | | Cum. rainfall ≥ 160 | 223 | 1180 | 1.9E-01 | 62.2 |
| | From 2020–10-02 06:00 to 2020–10-05 06:00 | Unaffected area | 1499 | 479289 | 3.1E-03 | 1.0 |
| | | Cum. rainfall ≥ 80 | 488 | 47489 | 1.0E-02 | 3.3 |
| | | Cum. rainfall ≥ 100 | 1006 | 18111 | 5.6E-02 | 17.8 |
| | | Cum. rainfall ≥ 160 | 333 | 3972 | 8.4E-02 | 26.8 |

earthquakes, the density of mentions is much higher in the areas where the seismic shakings are felt, with an amplification factor ranging from 15.1 for the earthquake of November 16, 2019, to 1734.7 for the earthquake of November 12, 2019. The latter case, which shows particularly strong activity around the epicenter in the city of Strasbourg, cannot however be explained only by the posts sent by direct witnesses of the earthquake. It seems to have "benefited" from particular attention for two reasons. Firstly, it occurred the day after the Le Teil earthquake, at a time when the French population was particularly attentive to the subject of seismic risk: indeed, a high level of activity is still being observed in the area affected by the Le Teil earthquake on November 12 and 13, 2019. In addition, the question of the human origin of the November 12 earthquake was quickly raised, with the hypothesis of a trigger due to geothermal pumping in the area: this led to the publication of numerous messages on Twitter. In the case of the heavy rainfall events of September and October 2020, the spatial density of mentions increases with very significant cumulative rainfalls. Thus, while areas affected by cumulative rainfalls ranging between 80 and 100 mm show no noticeable increase in the number of mentions, areas with values over 100 mm show, on the contrary, much higher densities of mentions than elsewhere, with an amplification factor varying between 17.8 and 78.5. These very marked trends reflect a good match between the spatial severity of events and the density of geolocated entities within tweets, which is particularly interesting for crisis managers.

Nonetheless, some aspects of our model need improvement. First, tweets are processed independently, hence preventing the model from building a macro-representation of the overall situation. This has a strong negative impact on ambiguous tweets and/or entities, where more contextual information is required. Let us consider the following two tweets as examples:

- "Tremblement de terre de Valence à Lyon. Vous l'avez senti aussi?" (English translation: "Earthquake from Valence to Lyon. Did you feel it too?")

- "Il vient d'y avoir un tremblement de terre à Valence" (English translation: "There has just been an earthquake in Valencia")

In French, "Valence" may refer to several municipalities located in France, or to Valencia, a city in Spain. Hence, in the original French version of the second tweet, it is not obvious which city is mentioned. However, in the first tweet, it is much more likely that "Valence" is referring to Valence in France because Lyon, one of the largest French cities, is also mentioned. As a consequence, our model did predict accurately the location of Valence in the first tweet, but not in the second. Such an error could have been avoided by remembering that previous tweets were talking about an earthquake in France, near Valence, instead of considering each tweet independently.

Switching to a multilingual model, instead of a pure French one, could also unlock new possibilities. The first obvious advantage would be to detect and geolocate any crisis without language limitation. But, more critically, this could enable the model to leverage all the available data in case of cross-border crises, such as the Alex storm that impacted both France and Italy. Our current model can process only French tweets, while many tweets have been posted by Italian people in distress.

## 7 Conclusion and perspectives

We propose a new bi-encoder architecture that simultaneously detects entity mentions in a document and links them to a knowledge base, instead of relying on two different models as commonly done. This allows us not only to fasten the entity-linking process but also to leverage existing NER annotated datasets to train our model to produce better representations while competing approaches are constrained to rely only on EL-annotated datasets. As such,

our approach is more suitable to deal with domain-specific datasets since it is not limited to being trained on the few EL datasets available.

We showed that training this architecture using publicly available data from Wikipedia in addition to a small specialized dataset allows efficient and accurate retrieval of locations mentioned in tweets. Beyond the general performance of the model to link the tweets to geolocated entities, it is clear with our qualitative evaluation that the model is able to capture the overall footprint of earthquakes and flash floods (thanks to messages exchanged by direct witnesses, but also because of related discussions), which is critical for crisis managers to build their common operational picture. While we focus, in this study, on the geolocation of information related to earthquakes and floods, our approach can be extended with minor changes to any crisis event such as, without limitation, forest fire, traffic accidents or terrorist attacks.

Our geolocation system can probably be improved by taking into consideration the state of the current situation. For instance, if an event has already been detected in a certain area, then this information should be used to improve the geolocation of future tweets. This could, for instance, be achieved by sharing a global situational embedding or by updating the attention mechanism of the Transformer model [76–78]. Furthermore, external expert knowledge, such as maps, could help the system to select the right candidates, for instance by favouring a city next to a river in case of flooding or a region known for its recurrent seismic activity to geolocate the damages caused by an earthquake.

We also plan to produce a multilingual version of this model in order to enable the monitoring of international crises, which is particularly critical to dealing with a cross-frontier crisis, such as, for example, the Alex storm that hit both France and Italy. It will require the annotation of tweets related to natural disasters in different languages.

Finally, the model actually links location mentions in the tweets with Wikidata features, but it is rare to get complex geometries such as polylines or polygons in Wikidata. A final step would be to link our entities to OpenStreetMap or another authoritative source of vector spatial data, via the Wikidata linking.

## Supporting information

**S1 File. CAp 2017 label mappings.**
(PDF)

**S2 File. HIPE-2022 label mappings.**
(PDF)

**S3 File. Twitter query for the Teil earthquake.**
(PDF)

**S4 File. Twitter query for the Alex storm.**
(PDF)

## Acknowledgments

The authors would like to thank F. Boulahya, Y. Retout, C. Mato, A. Montarnal, B. Farah and F. Smai for their help in annotating data, as well as the MAIF Foundation for its support to the design and development of the SURICATE-Nat platform. We also thank BCSF for giving us access to its French macroseismic data distribution webservice, and L. Bernede from Météo France for providing us with the rainfall data recorded during Alex storm.

## Author Contributions

**Conceptualization:** Gaëtan Caillaut, Cécile Gracianne, Nathalie Abadie, Guillaume Touya.

**Data curation:** Gaëtan Caillaut.

**Funding acquisition:** Samuel Auclair.

**Methodology:** Gaëtan Caillaut, Nathalie Abadie.

**Project administration:** Samuel Auclair.

**Supervision:** Cécile Gracianne, Nathalie Abadie, Guillaume Touya.

**Validation:** Samuel Auclair.

**Visualization:** Gaëtan Caillaut, Samuel Auclair.

**Writing – original draft:** Gaëtan Caillaut, Samuel Auclair, Guillaume Touya.

**Writing – review & editing:** Cécile Gracianne, Nathalie Abadie.

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
