## [Decision Letter · Decision Letter 0]

11 Apr 2023

PONE-D-23-07957Entity Linking for real-time geolocation of natural disasters from social network postsPLOS ONE

Dear Dr. Caillaut,

Thank you for submitting your manuscript to PLOS ONE. After careful consideration, we feel that it has merit but does not fully meet PLOS ONE’s publication criteria as it currently stands. Therefore, we invite you to submit a revised version of the manuscript that addresses the points raised during the review process. Please submit your revised manuscript by May 26 2023 11:59PM. If you will need more time than this to complete your revisions, please reply to this message or contact the journal office at plosone@plos.org. Please include the following items when submitting your revised manuscript:A rebuttal letter that responds to each point raised by the academic editor and reviewer(s). You should upload this letter as a separate file labeled 'Response to Reviewers'.A marked-up copy of your manuscript that highlights changes made to the original version. You should upload this as a separate file labeled 'Revised Manuscript with Track Changes'.An unmarked version of your revised paper without tracked changes. You should upload this as a separate file labeled 'Manuscript'.

We look forward to receiving your revised manuscript.

Kind regards,

Baby Gobin

Academic Editor

PLOS ONE

Journal Requirements:

2. In your Methods section, please include additional information about your dataset and ensure that you have included a statement specifying whether the collection and analysis method complied with the terms and conditions for the source of the data.

   "The work presented in this article was carried out with funding from the Agence Nationale de la Recherche (ANR) within the framework of the CARNOT institutes, as well as within the the R´eSoCIO project co-funded by ANR under the grant ANR-20-CE39-001. Opinions expressed in this paper solely reflect the authors’ view; the ANR is not responsible for any use that may be made of information it contains. The authors would like to thank F. Boulahya, Y. Retout, C. Mato, A. Montarnal, B.Farah and F. Smai for their help in annotating data, as well as the MAIF Foundation for its support to the design and development of the SURICATE-Nat platform. We also thank BCSF for giving us access to its French macroseismic data distribution webservice, and L. Bernede from M´et´eo France for providing us with the rainfall data recorded during Alex storm."

 "The work presented in this article was carried out with funding from the French Research Agency (ANR - https://anr.fr/en/) within the framework of the CARNOT institutes (Gaëtan Caillault), as well as within the the RéSoCIO project co-funded by ANR under the grant ANR-20-CE39-001 (Samuel Auclair & Cécile Gracianne). Opinions expressed in this paper solely reflect the authors' view; the ANR is not responsible for any use that may be made of information it contains.

6. We note that Figure 4 and 6 in your submission contain [map/satellite] images which may be copyrighted. All PLOS content is published under the Creative Commons Attribution License (CC BY 4.0), which means that the manuscript, images, and Supporting Information files will be freely available online, and any third party is permitted to access, download, copy, distribute, and use these materials in any way, even commercially, with proper attribution. For these reasons, we cannot publish previously copyrighted maps or satellite images created using proprietary data, such as Google software (Google Maps, Street View, and Earth). For more information, see our copyright guidelines: http://journals.plos.org/plosone/s/licenses-and-copyright.

a. You may seek permission from the original copyright holder of Figure 4 nd 6 to publish the content specifically under the CC BY 4.0 license.  

6. We note you have included a table to which you do not refer in the text of your manuscript. Please ensure that you refer to Table 5 in your text; if accepted, production will need this reference to link the reader to the Table.

Additional Editor Comments:

You are required to update your manuscript based on the comments of the reviewers . 

Reviewers' comments:

Reviewer's Responses to Questions

**Comments to the Author**

1. Is the manuscript technically sound, and do the data support the conclusions?

Reviewer #1: Yes

Reviewer #2: Partly

2. Has the statistical analysis been performed appropriately and rigorously? 

Reviewer #1: Yes

Reviewer #2: No

3. Have the authors made all data underlying the findings in their manuscript fully available?

Reviewer #1: No

Reviewer #2: No

4. Is the manuscript presented in an intelligible fashion and written in standard English?

Reviewer #1: Yes

Reviewer #2: Yes

5. Review Comments to the Author

Reviewer #1: This paper discusses the challenges of using social networks to develop situational awareness during natural disasters and proposes a method for detecting and geolocating French messages on Twitter to build maps in real-time. The authors demonstrate that their system performs as well as state-of-the-art systems and can contribute to automatic social network analysis for crisis managers. In my opinion, this work turns out to be technically valid even if it shows limitations as specified below:

- Novelty aspects: explain the novelty aspects of the paper better, also through concrete examples. The paper seems to be not very innovative, i.e. the application of known techniques adapted to the French context.

- Related work (1): For each manuscript cited the main differences with the proposed approach must be highlighted. A comparative table could help the readers to understand the differences among the different works present in the literature and the strengths of this work.

- Related work (2): The paper "Using Social Media for Sub-Event Detection during Disasters" it turns out to be a related work very close to the one proposed. In particular, it proposes a technique for identifying the sub-events that occur after a disaster. How is your work different?

- Dataset and code: To allow the reproducibility of the experiments it is necessary to publish the datasets on a public repository and share the link in the paper. Also, the method code should be made public to allow experiments to be reproduced and the results obtained to be validated.

- Experiments: Since the authors of this paper have a dataset with labels, why didn't you carry out quantitative analyses on the accuracy of your method (F1-score) in detecting events?

Reviewer #2: In this paper, the task of detecting and geolocating information in French language is addressed. The contributions are a data set and an entity-linking pipeline. Due to the following reasons, I would like to recommend that the paper needs significant revisions before publishing:

- the proposed models need more detailed descriptions

- the narrative of the paper needs to be enhanced to ensure that the reader can clearly see "the plot"

- the task is not well described and changes from EL (place mentions in texts) to geolocation of natural disasters - the latter is definitely not in scope of the involved methods as single tweets are processed (geo-located)

- the training data itself is partially generated from ML models, of which the actual performance is unclear - this needs to be further investigated

- more quantitative experiments related to the proposed models are missing but definitely required: how well is the NER-step working? how do errors propagate? what is the effect of the cross-encoder?

- an interpretation of the scores in table 6 is missing: what are reasons for the poor F1-scores?

- while also highlighted in the manuscript title, the real-time aspect is not addressed at all

- Why are entities involved, which will actually have not coordinate (e.g. person)?

More detailed comments:

### Abstract

"show that despite these additional constraints"  not sure to which constraints is referred here?

### Introduction

32/33 "one of the main issues of these automatic analyses is the ability to correctly place the information extracted  in a map" - might be, but not the first one - how about overload reduction?

While reading the first two chapters, a question comes up: what is the actual motivation of entity linking here? Why do the authors expect a benefit for the task (hypothesis)? It may be clear or the reader might have an idea, but reading the author's perspective would give more insights here.

61: "we propose a pipeline to automatically geolocate natural disasters from tweets"  this sentence suggests that events will be detected, but the task is related to single tweets, when I get it right. Hence, geolocating "natural disasters" might not be the appropriate term here.

Would actually be good to also read from the obtained results / performance abilities of the proposed method in the introduction

### Task description and related works

I like that we can find a description of the task first. It may be good to provide a list of examples to explicitly (1) show the different types of toponyms, the authors are actually addressing, and (2) the target types of place description (points, polygons,...?). Later (2.2), we can find

The task "mentioned location prediction" is also known as geoparsing.

"but we will focus more on methods addressing a similar crisis management context" I can understand this approach, but sometimes it is worth doing a review independently from a domain - it may turn out that there are approaches around that are not well known in disaster management but worth to test? Just a thought..

In line with reference 26, this one might also be of interest: https://ieeexplore.ieee.org/abstract/document/9711571

In addition to the mentioned approaches in the related work, it would be good to also read about the open issues / drawbacks they have - at least those that are addressed in this paper.

While reading 2.2, I am wondering why the description of the EL task is not part of the introduction of the chapter (where the task is described)? This would give a comprehensive overview of all involved sub-tasks that are addressed in this work followed by related work. In turn, chapter 2.2 focuses a lot on the task and only a few related works are mentioned. I would recommend to re-structure this chapter.

134: The meaning of BIO could be explained.

156: dated  outdated?

180: "Detecting and geolocating natural disasters"  this suggests that events are detected. However, this is not in line with the task description, which works on a document-level

3.1: Here, the first part of the model is described. For better understandability, I would favor to get a first overarching overview of the proposed method followed by the detailed descriptions.

Figure 1: In its current form, the figure does not contain all relevant information. For instance, what do the green vectors/matrices contain/represent? How is the final embedding actually is computed?

211: "At inference time, entity embeddings can be pre-computed,": This statement is a bit confusing, as the embeddings for the entities are already pre-computed (i.e., at inference-time, only the mention embedding has to be computed)? Which metric is actually used to compare embedding vectors?

217ff: "Then, we propose to rely on the first token, instead of the CLS token, of an entity mention to produce mention embeddings, thus allowing to embed all the entity mentions of a document at the same time."  It would then be required to explain, how multiple mentions - especially the expected varying number - is handled

218: "we propose to rely on the first token": why not B and I considered (as some mentions might consist of more than 1 or 2 tokens)? OK, addressed in line 225..

3.2 Cross-Encoder: The description is not quite detailed and could be enhanced. How does it eventually help to mitigate the aforementioned potential errors?

288: "the best performances are obtained with": this statement seems to be quite holistic - I would rather say that this is true according to the conducted experiments in the mentioned paper?

324: "beforehand, The" -> the

342ff: When I see the list of annotation labels considered, I am a bit confused. Taking into account types that do not describe places differs from the task?

360: "We then applied this classifier to classify all the French Wikipedia entities ": This tends to be critical, as the labels will potentially not have the quality of manual annotations. It would be required to read about test results using unseen test data.

385ff: These two sentences are representative for the writing style of the whole article, where the order of information presented appears to be a bit confusing. I just had a look the the paper "Entity Linking in 100 Languages" - the writing style is very crisp, clear and structured. I would recommend to revise the manuscript to achieve that readers can follow a bit easier.

4.3: It is not clear, how the data is used when the EL-related part is missing?

400: "it then appears to be extremely valuable to help the model build coherent representations for tweets"  this statement needs to be confirmed.

406: "geolocate natural disasters from social networks": well, event if this is the application, it is still about place mentions, right? I would say that it of course depends on the involved document types - if Twitter is the target platform, a model needs to be trained with data from Twitter. But it might not be necessary to use disaster tweets to detect and link the place names. Might be a good idea for an experiment?

Table 5 seems not to be mentioned or referenced in the text.

Table 6: EL performance is shown here. It would also be interesting to see, how well the mentions actually are detected as this would directly influence the linking. A (maybe stupid) question related to this: what would happen, if we simply take the NER results search the entities with these keywords (maybe allowing for some letter permutations)?

473: "given in Appendix 7 and 7" 7 appears twice

493: "Since our model detects non-geographical entities too": this rises the question, why the other types are then actually still included? Doesn't this make the task more complex for the models?

Table 7: "The number between parenthesis indicates the number of tweets, mentions or entities which have been localized inside the area impacted by the earthquake/storm.": as there are around 250 distinct entities found, it would be of interest (and feasible in terms of effort) to know the quality here (i.e., false positive rates or other metrics?).

498: EMSC should be explained

548: "While this last analysis does not prove that our model predictions are correct".. this was exactly what I was thinking here: I would at first be interested in how well the proposed models actually perform - this is necessary. Another aspect: it is well known, that keyword-based filtering of tweets comes at the cost of a bad precision. How is ensured, that non-related tweets are not used in this analysis?

6.2: As the title is "Alex storm", it is a bit surprising/unexpected to read about floods in 6.2.2?

598 (Discussion): "Our model seems to be able to capture coherent representations of real natural disasters."  very vague - has to be underpinned with quantitative experiments

638: "Without being able to assess in detail the capacity of the model to detect all the geo-locatable features with precision": why not taking the usual approach of random samples?

640: "the model is able to capture the overall footprint of earthquakes and flash floods": the proposed model can actually do EL and since the linked entities have geo-coordinates, they can be shown in a map. The fact that a footprint is visible, is related to the data and the users that report on an event. however, it is not clear, if the data contains many eyewitness-reports or (as commonly observed) sympathy/support messages. as the tweets are basically identified based on keywords, it is likely, that there are many false positives contained. this needs to be investigated based on quantitative and qualitative experiments.

650: "Furthermore, external expert knowledge, such as maps,.." this is actually a crucial point. no emergency response manager will solely decide based on a twitter system. they in fact already have well-established routines and have local knowledge so that we need to identify the information gaps

6. PLOS authors have the option to publish the peer review history of their article (what does this mean?). If published, this will include your full peer review and any attached files.

Reviewer #1: No

Reviewer #2: No

---

## [Author Response · Author response to Decision Letter 0]

7 Dec 2023

All comments and responses to the reviewers can be found in the "response_to_reviewers.pdf" file included.

---

## [Decision Letter · Decision Letter 1]

5 Mar 2024

PONE-D-23-07957R1Entity Linking for real-time geolocation of natural disasters from social network postsPLOS ONE

Dear Dr. Caillaut,

Thank you for submitting your manuscript to PLOS ONE. After careful consideration, we feel that it has merit but does not fully meet PLOS ONE’s publication criteria as it currently stands. Therefore, we invite you to submit a revised version of the manuscript that addresses the points raised during the review process.

The provided revision of the paper is valid and overall of good quality, however the reviewers have raised some further points to address which I believe would improve the manuscript and may allow a revised version to reach an acceptable level for the publication of the paper.

You are then required to update your manuscript based on the comments of the reviewers.

We look forward to receiving your revised manuscript.

Kind regards,

Sergio Consoli

Academic Editor

PLOS ONE

Reviewers' comments:

Reviewer's Responses to Questions

**Comments to the Author**

1. If the authors have adequately addressed your comments raised in a previous round of review and you feel that this manuscript is now acceptable for publication, you may indicate that here to bypass the “Comments to the Author” section, enter your conflict of interest statement in the “Confidential to Editor” section, and submit your "Accept" recommendation.

Reviewer #1: All comments have been addressed

Reviewer #3: All comments have been addressed

Reviewer #4: (No Response)

2. Is the manuscript technically sound, and do the data support the conclusions?

Reviewer #1: Yes

Reviewer #3: Yes

Reviewer #4: Yes

3. Has the statistical analysis been performed appropriately and rigorously? 

Reviewer #1: Yes

Reviewer #3: I Don't Know

Reviewer #4: No

4. Have the authors made all data underlying the findings in their manuscript fully available?

Reviewer #1: No

Reviewer #3: Yes

Reviewer #4: Yes

5. Is the manuscript presented in an intelligible fashion and written in standard English?

Reviewer #1: Yes

Reviewer #3: Yes

Reviewer #4: Yes

6. Review Comments to the Author

Reviewer #1: The authors, while providing concise responses to the reviewers' inquiries, have successfully enhanced the paper. Consequently, I recommend accepting the submission.

Reviewer #3: In general, the paper looks well-worked-upon and useful; I need to especially praise the authors for their willing to improve a concrete practice, rather than elaborate on a method that will never be actually used in real life.

I am not questioning the general approach; neither I am against the method in its major elements (design, pipeline, evaluation etc.). I appreciate the work with the datasets; they include nearly all available relevant data, as well as the new training dataset created by the authors.

But I have several reservations that, to my viewpoint, need to be addressed, to make the paper less ‘technical’ and a bit closer to real life, including the practical disaster fighting by both professionals and ordinary people. These include conceptual issues and issues of methodology.

CONCEPTUAL ISSUES

1. The information on disasters is usually gathered not from social media (which may be highly misleading, as no fact-checking is done), but by special taskforces. Thus, why use your tool instead of using the fact-checked evidence collected by professionals? And what to do when the data from the former contradict the latter? What would your advice be? I think this issue needs to be mentioned in the paper, as stating unequivocally that such a tool only helps may, too, be misleading. We know from many examples, even before Twitter, e.g. from fires in Greece and other European countries that people tended to listen to media that reported on fires, not the firemen, and came to wrong places for help. Social media are an even more distorted mirrors of events; they may intensify, distort, misrepresent, and diminish the scale of the event. Thus, it needs to be clarified in the paper how exactly the tool is used and what for, knowing that social media may mislead the viewer, and how the info is doublechecked; this will explain why and how you wish to enhance its work.

2. However, even if your tool will represent the social media reality 100%, there is no guarantee that it corresponds 100% to what is happening on the ground. This, in its turn, means that the uses of the tool need to be limited to, say, disaster alerts – or, on the opposite, the tool needs to be ML-enhanced to be able to go beyond naïve search for locations and keywords but get trained to detect markers of both disasters and locations beyond particular names/keywords, in order to critically improve its quality.

METHODS AND RESULTS

3. The authors are right in saying that there are too many papers on Twitter geolocation; but I am not sure they are right in disregarding all those that tackle issues beyond natiral disaster detection. This is especially wrong in terms of methodology: Many papers that do not focus on monitoring disasters still provide valuable improvements for the detection method itself. A short review may be found in Blekanov et al. (2022) – see https://dl.acm.org/doi/abs/10.1007/978-3-031-22131-6_2. As an example of what is omitted by the authors, this paper itself might also be of help, as it compares three approaches to enhance geolocation of Twitter users, including via ML-based analysis of their text corpora, and reports improving user geolocation. I am not sure how it will work on the community/local level, but it worked well on national level, as the authors claim.

4. In connection with (4): Analysing individual tweets may be productive for ‘red alerts’, but may remain endlessly non-productive for geolocation detection, however good annotation, model training, and fine-tuning may be. As a ‘future work’ prospect, I would still mention analysis of tweet pools. You tell that parsing the history of the network is unfeasible when you discuss such options, and claim you do not use prior knowledge in your model; but this is not about parsing the whole network but only the tweets of a particular account, of which the tweets would allow for quicker recognition of the residence of a given user who posted the info on a disaster. The same goes from the friendship network, but here the data may be more misleading (e.g., if one would detect my location based on my Facebook feed, they would be wrong by 4,400 km).

5. Low Recall and F-score are not explained (I think the reviewers had already mentioned that). Either explain why this result is sufficient and cannot be higher (is it so for tweets because of their length?) or at least highlight the results that you orient to and explain why recall and F-score are not that important (which I honestly doubt).

Reviewer #4: This manuscript presents an entity-linking pipeline for geolocating French Tweets posted during natural disasters. Furthermore, the manuscript introduces an annotated entity-linking data set consisting of French Wikipedia corpus and French tweets collected during major disasters. I would recommend accepting this paper after clarifying a few points (elaborated below).

Strengths:

• The qualitative evaluation is good, demonstrating the utility of the proposed model for disaster response.

• The paper is well-written. The organization of the paper could be improved, but it remains easy to follow.

• The creation of an annotated data set that can be used for disaster response Named Entity Recognition and Entity Linking tasks in other languages is a valuable contribution, given that the mainstream research in this area is focusing on English data.

Concerns:

• The authors should clarify whether the proposed entity-linking pipeline presented in this manuscript presents a novel contribution. Are there similar entity-linking models that have been proposed to address this specific issue? Is the proposed architecture unique? A comparison with previous works is recommended.

• The cross-encoder component in the proposed pipeline is not clear. Clarifications are needed on whether this cross-encoder was trained separately or on the output of the bi-encoder. It will be helpful to add a figure illustrating all the pipeline’s components and how they are linked to each other’s.

• The categories of 'RISKNAT' and 'DAMAGES' (page 11) are unclear. Are they location-related categories? How do they differ from the 'GEOLOC' category? The authors provided a few examples of the 'RISKNAT' category in page 17 “information describing the phenomenon” such as “heavy rain” and “strong gusts of wind”, which do not seem to be location related. It would be beneficial if the authors explain more what these two categories are referring to in page 11.

• In section 5.2 (Results), the authors stated “Recently, a comparison of several NER models showed that state-of-the-art F1 scores are smaller than 75% [64], meaning that the results of the Bi-Encoder are excellent”. It is not clear whether the F1 scores mentioned here are those trained and tested on French data. If it is the case, I think the F1 score obtained on French and English data are not directly comparable. There might be some variations in the way locations are mentioned in these two languages. Even though the scope of this paper is French tweets, I would suggest training and testing the NER component of the proposed pipeline on an existing English Benchmark (e.g., the one proposed in “Suwaileh R, Elsayed T, Imran M, Sajjad H. When a disaster happens, we are ready: Location Mention Recognition from crisis tweets. International Journal of Disaster Risk Reduction. 2022; p. 103107”) and compare its performance to the state-of-the-art NER models to support the above conclusion.

7. PLOS authors have the option to publish the peer review history of their article (what does this mean?). If published, this will include your full peer review and any attached files.

Reviewer #1: No

Reviewer #3: No

Reviewer #4: **Yes: **Ghaith Rabadi

---

## [Author Response · Author response to Decision Letter 1]

15 Apr 2024

We would to thank the reviewers and the editor for their useful additional comments. We took their recommendations into account and answered their questions. Our answers to all comments can be found in red below.

Reviewer #3: In general, the paper looks well-worked-upon and useful; I need to especially praise the authors for their willing to improve a concrete practice, rather than elaborate on a method that will never be actually used in real life.

I am not questioning the general approach; neither I am against the method in its major elements (design, pipeline, evaluation etc.). I appreciate the work with the datasets; they include nearly all available relevant data, as well as the new training dataset created by the authors.

But I have several reservations that, to my viewpoint, need to be addressed, to make the paper less ‘technical’ and a bit closer to real life, including the practical disaster fighting by both professionals and ordinary people. These include conceptual issues and issues of methodology.

CONCEPTUAL ISSUES

 1. The information on disasters is usually gathered not from social media (which may be highly misleading, as no fact-checking is done), but by special taskforces. Thus, why use your tool instead of using the fact-checked evidence collected by professionals? And what to do when the data from the former contradict the latter? What would your advice be? I think this issue needs to be mentioned in the paper, as stating unequivocally that such a tool only helps may, too, be misleading. We know from many examples, even before Twitter, e.g. from fires in Greece and other European countries that people tended to listen to media that reported on fires, not the firemen, and came to wrong places for help. Social media are an even more distorted mirrors of events; they may intensify, distort, misrepresent, and diminish the scale of the event. Thus, it needs to be clarified in the paper how exactly the tool is used and what for, knowing that social media may mislead the viewer, and how the info is doublechecked; this will explain why and how you wish to enhance its work.

Reply: Thank you for this very interesting comment. First of all, it is very true to point out that monitoring on social networks in support of crisis management is often carried out manually by teams of volunteers. Our aim is therefore not to replace these teams, but rather to support them by developing automatic processing tools allowing them to increase efficiency. The value of human analysis remains very important, especially to verify the extracted information. A paragraph has been added to this effect in the introduction.

 2. However, even if your tool will represent the social media reality 100%, there is no guarantee that it corresponds 100% to what is happening on the ground. This, in its turn, means that the uses of the tool need to be limited to, say, disaster alerts – or, on the opposite, the tool needs to be ML-enhanced to be able to go beyond naïve search for locations and keywords but get trained to detect markers of both disasters and locations beyond particular names/keywords, in order to critically improve its quality.

Reply: We fully agree with the reviewer that social networks should not be used alone to build situational awareness, at the risk of getting a distorted vision of the reality. This is addressed in the introduction, which points out that social networks must be considered "in addition to traditional actionable channels". Furthermore, we believe that extracting geolocation information is a way of complementing naïve search with advanced analysis, enabling for example the detection of spatio-temporal anomalies. However, how this information extracted from messages can be used to infer higher-level actionable intelligence is outside the scope of this article.

METHODS AND RESULTS

 3. The authors are right in saying that there are too many papers on Twitter geolocation; but I am not sure they are right in disregarding all those that tackle issues beyond natiral disaster detection. This is especially wrong in terms of methodology: Many papers that do not focus on monitoring disasters still provide valuable improvements for the detection method itself. A short review may be found in Blekanov et al. (2022) – see https://dl.acm.org/doi/abs/10.1007/978-3-031-22131-6_2. As an example of what is omitted by the authors, this paper itself might also be of help, as it compares three approaches to enhance geolocation of Twitter users, including via ML-based analysis of their text corpora, and reports improving user geolocation. I am not sure how it will work on the community/local level, but it worked well on national level, as the authors claim.

Reply: Works on the geolocation of tweets may indeed be of methodological interest for our purpose, even if they do not deal directly with natural disasters. We were primarily interested in analyses of tweets sent during natural disasters, as these have the triple advantage of presenting not only the geolocation approaches implemented, but also the datasets used and the evaluation methodologies adopted for this particular use case. This enabled us to build a training corpus adapted to the texts to be processed, and to propose evaluation measures appropriate to the level of detail of the mentions of named spatial entities to be localized.

The approach proposed in the article referred to remains interesting in our case: thank you for bringing this to our attention. This work is based on the same steps as the ones we already present in the related woks section: here, the aim is to use a deep neural network-based NER model to recognize places mentioned in tweets (provided by the Spacy library), and then to associate the recognized mentions with coordinates using a geocoding tool (OSM Nominatim). This second step involves a linking operation between each extracted mention of a named place and a knowledge base of reference locations. A graph of users’ interactions is added to improve the named place disambiguation process. This is in line with approaches presented in our state of the art, such as that of (Jurgens, 2013), which also uses the relationship graph of twitter users.

The approach we propose differs from that presented in this article in that it performs these two successive tasks in a single step. This has the advantage of allowing the textual context of spatial named entities, rich in spatial relationships with neighboring named places, to be taken into account during the linking stage, thus facilitating the disambiguation of candidate places. We have therefore added a reference in the article to clarify our positioning in relation to this kind of approaches.

 4. In connection with (4): Analysing individual tweets may be productive for ‘red alerts’, but may remain endlessly non-productive for geolocation detection, however good annotation, model training, and fine-tuning may be. As a ‘future work’ prospect, I would still mention analysis of tweet pools. You tell that parsing the history of the network is unfeasible when you discuss such options, and claim you do not use prior knowledge in your model; but this is not about parsing the whole network but only the tweets of a particular account, of which the tweets would allow for quicker recognition of the residence of a given user who posted the info on a disaster. The same goes from the friendship network, but here the data may be more misleading (e.g., if one would detect my location based on my Facebook feed, they would be wrong by 4,400 km).

Reply: We purposely chose not to analyze individuals for three main reasons. First, we aim to geolocate the events mentioned in the social media post. As you said, user data is not fully trustable, but even if it were, the user’s home location can be misleading for many reasons as people often refers to event far from where they live, or they can be away for vacation…

The second reason is that we did not want to actively “spy” on users, for ethical and RGPD reasons. Even if we don’t store user information, we are not sure of the legal implication on such data collection, so we preferred to avoid it.

Lastly, there are technical reasons. The twitter API is quite limiting when it comes to link mining. So, looking at users’ friend network would quickly exhaust our allowed API calls.

Yet, we agree, one should be able to get better results by gathering network information, that’s why we did explore this path, but we let it out to future works for these particular reasons. 

 5. Low Recall and F-score are not explained (I think the reviewers had already mentioned that). Either explain why this result is sufficient and cannot be higher (is it so for tweets because of their length?) or at least highlight the results that you orient to and explain why recall and F-score are not that important (which I honestly doubt).

Reply: You are right, we should have discussed the F-1 score for the NER task. We did not in the previous version because we consider that it is not a low score but rather a good score. NER in tweets is a very complex task. If we look at current benchmarks for NET in tweets (e.g. https://paperswithcode.com/dataset/wnut-2016-ner), the recall and F1 scores are even lower due to the properties of tweets: they are short and often written with shorthands and abbreviations. For instance, in our dataset, we find a tweet that can be translated as “the ground shakes in stras” where “stras” means Strasbourg, the city in the east of France. Those instances are very complex to recognize. We added a similar discussion in the text to explain those values.

Reviewer #4: This manuscript presents an entity-linking pipeline for geolocating French Tweets posted during natural disasters. Furthermore, the manuscript introduces an annotated entity-linking data set consisting of French Wikipedia corpus and French tweets collected during major disasters. I would recommend accepting this paper after clarifying a few points (elaborated below).

Strengths:

• The qualitative evaluation is good, demonstrating the utility of the proposed model for disaster response.

• The paper is well-written. The organization of the paper could be improved, but it remains easy to follow.

• The creation of an annotated data set that can be used for disaster response Named Entity Recognition and Entity Linking tasks in other languages is a valuable contribution, given that the mainstream research in this area is focusing on English data.

Concerns:

• The authors should clarify whether the proposed entity-linking pipeline presented in this manuscript presents a novel contribution. Are there similar entity-linking models that have been proposed to address this specific issue? Is the proposed architecture unique? A comparison with previous works is recommended.

Reply: The novelty of this approach comes primarily from (1) the use of entity-linking to perform geolocation and (2) the updates made on the bi-encoder to simultaneously retrieve and classify all mentions in a text. It does not really change the way people do entity-linking, as the principles remain the same (retrieve then rerank), but it does improve the efficiency of exisisting systems by a lot.

We made these contribution clearer both in the introduction and Section 3.

There are other minor novelties, such as the training set used to train our cross-encoder.

• The cross-encoder component in the proposed pipeline is not clear. Clarifications are needed on whether this cross-encoder was trained separately or on the output of the bi-encoder. It will be helpful to add a figure illustrating all the pipeline’s components and how they are linked to each other’s.

Reply: All models are trained separately. However, the cross-encoder depend on the bi-encoder, as it is trained on data generated by the bi-encoder. We added more details in the paper, but the general idea is that we need to train the cross-encoder on realistic data. During inference, the cross-encoder will be asked to rerank candidates retrieved by the bi-encoder, but they will (should) be very similar to each others. Reranking similar objects is a very difficult task, so we need to introduce this complexity during the training of the model. That’s why we cannot just randomly sample negative example to train the model. We need to actively mine them. We can do this manually, but it’s easier and more maintainable to generate them automatically using the trained bi-encoder.

• The categories of 'RISKNAT' and 'DAMAGES' (page 11) are unclear. Are they location-related categories? How do they differ from the 'GEOLOC' category? The authors provided a few examples of the 'RISKNAT' category in page 17 “information describing the phenomenon” such as “heavy rain” and “strong gusts of wind”, which do not seem to be location related. It would be beneficial if the authors explain more what these two categories are referring to in page 11.

Reply: The categories of 'RISKNAT' and 'DAMAGES' clearly differ from the 'GEOLOC' one, as they do not contain any geographic information. These categories are considered here because we consider them important for the construction of a "common operational picture": in particular, their geospatial analysis (via inferred geolocation) seems very relevant, for example to identify highly impacted sectors (e.g. spatial concentration of messages associated with the 'DAMAGES' category) or natural induced effects such as landslides (e.g. spatial concentration of messages associated with the 'RISKNAT' category). The explanation of these two categories in page 11 has been modified to be more explicit.

• In section 5.2 (Results), the authors stated “Recently, a comparison of several NER models showed that state-of-the-art F1 scores are smaller than 75% [64], meaning that the results of the Bi-Encoder are excellent”. It is not clear whether the F1 scores mentioned here are those trained and tested on French data. If it is the case, I think the F1 score obtained on French and English data are not directly comparable. There might be some variations in the way locations are mentioned in these two languages. Even though the scope of this paper is French tweets, I would suggest training and testing the NER component of the proposed pipeline on an existing English Benchmark (e.g., the one proposed in “Suwaileh R, Elsayed T, Imran M, Sajjad H. When a disaster happens, we are ready: Location Mention Recognition from crisis tweets. International Journal of Disaster Risk Reduction. 2022; p. 103107”) and compare its performance to the state-of-the-art NER models to support the above conclusion.

Reply: You are completely right about the NER scores, they correspond to benchmarks with English tweets, and there is no theoretical guarantee that the task is similarly complex in French. However, we made this assertion based on recent research showing that there is no major difference in the NER task between English and French, and other languages (https://doi.org/10.1016/j.artint.2012.03.006). We added this reference to the manuscript to explain that the NER score in French can be compared to other NER scores in English.

Regarding your suggestion to test on NER benchmarks, such as the ones used in the cited paper, it is indeed a very good idea. However, as explained in our previous round of reviews, the first author has now left our organisations, and he does not have enough time now with his new job to conduct these new experiments. For the other authors, it would be much more time consuming to carry out these experiments. This is why we decided not to follow your suggestion, and we hope you will not hold it against us.

---

## [Decision Letter · Decision Letter 2]

3 Jul 2024

Entity Linking for real-time geolocation of natural disasters from social network posts

PONE-D-23-07957R2

Dear Dr. Caillaut,

We’re pleased to inform you that your manuscript has been judged scientifically suitable for publication and will be formally accepted for publication once it meets all outstanding technical requirements.

Kind regards,

Sergio Consoli

Academic Editor

PLOS ONE

Additional Editor Comments (optional):

Reviewers' comments:

Reviewer's Responses to Questions

**Comments to the Author**

1. If the authors have adequately addressed your comments raised in a previous round of review and you feel that this manuscript is now acceptable for publication, you may indicate that here to bypass the “Comments to the Author” section, enter your conflict of interest statement in the “Confidential to Editor” section, and submit your "Accept" recommendation.

Reviewer #4: All comments have been addressed

2. Is the manuscript technically sound, and do the data support the conclusions?

Reviewer #4: Yes

3. Has the statistical analysis been performed appropriately and rigorously? 

Reviewer #4: Yes

4. Have the authors made all data underlying the findings in their manuscript fully available?

Reviewer #4: Yes

5. Is the manuscript presented in an intelligible fashion and written in standard English?

Reviewer #4: Yes

6. Review Comments to the Author

Reviewer #4: The authors have sufficiently addressed my previous comments. They highlighted the novelty of the proposed approach, which is an acceptable contribution worth of publishing.

The experiments were conducted using a manually annotated data set comprising 339 tweets. The small size of the data set used for evaluation is a main limitation of this research.

The authors have provided a justification for the direct comparison of their NER on French data against the state-of-the-art English NER results, which I find satisfactory at this stage. This presents a promising direction for future work, particularly in understanding how NER works across different languages, aiming to develop cross-language NER models.

Overall, the manuscript was well-written and clear, with no major issues identified.

7. PLOS authors have the option to publish the peer review history of their article (what does this mean?). If published, this will include your full peer review and any attached files.

Reviewer #4: **Yes: **Ghaith Rabadi

---

## [Editor Report · Acceptance letter]

12 Jul 2024

PONE-D-23-07957R2 

PLOS ONE

Dear Dr. Caillaut, 

I'm pleased to inform you that your manuscript has been deemed suitable for publication in PLOS ONE. Congratulations! Your manuscript is now being handed over to our production team.

Kind regards, 

on behalf of

Dr. Sergio Consoli 

Academic Editor

PLOS ONE